# The Zone of Influence: Matching sea level variability from coastal altimetry and tide gauges for vertical land motion estimation

Julius Oelsmann[1], Marcello Passaro[1], Denise Dettmering[1], Christian Schwatke[1], Laura Sanchez[1], and Florian Seitz[1]

[1]Deutsches Geodätisches Forschungsinstitut der Technischen Universität München, Arcisstraße 21, 80333 Munich, Germany

**Correspondence:** Julius Oelsmann (julius.oelsmann@tum.de)

**Abstract.**

Vertical land motion (VLM) at the coast is a substantial contributor to relative sea level change. In this work, we present a refined method for its determination, which is based on the combination of absolute satellite altimetry (SAT) sea level measurements and relative sea level changes recorded by tide gauges (TG). These measurements complement VLM estimates from GNSS (Global Navigation Satellite System) by increasing their spatial coverage. Trend estimates from SAT and TG combination are particularly sensitive to the quality and resolution of applied altimetry data as well as to the coupling procedure of altimetry and TGs. Hence, a multi-mission, dedicated coastal along-track altimetry dataset is coupled with high-frequency TG measurements at 58 stations. To improve the coupling-procedure, a so-called 'Zone of Influence' (ZOI) is defined, which confines coherent zones of sea level variability on the basis of relative levels of comparability between TG and altimetry observations. Selecting 20% of the most representative absolute sea level observations in a 300 km radius around the TGs results in the best VLM-estimates in terms of accuracy and uncertainty. At this threshold, $VLM_{SAT-TG}$ estimates have median formal uncertainties of 0.58 mm/year. Validation against GNSS VLM estimates yields a root-mean-square ($RMS_{\Delta VLM}$) of $VLM_{SAT-TG}$ and $VLM_{GNSS}$ differences of 1.28 mm/year, demonstrating the level of accuracy of our approach. Compared to a reference 250 km radius selection, the 300 km Zone of Influence improves trend accuracies by 15% and uncertainties by 35%. With increasing record lengths, the spatial scales of the coherency in coastal sea level trends increase. Therefore the relevance of the ZOI for improving $VLM_{SAT-TG}$ accuracy decreases. Further individual Zone of Influence adaptations offer the prospect of bringing the accuracy of the estimates below 1 mm/year.

## 1 Introduction

Coastal vertical land motion (VLM) significantly contributes to relative sea level change (SLC). VLM is in many places of the same order of magnitude (1-10 mm/year) as the sea level rise signal itself and displays significant spatial variations (Santamaría-Gómez et al., 2012). Consequently, VLM affects coastal impacts of climate-sensitive processes and can regionally account for large fractions of the observed and projected coastal SLC signal (Wöppelmann and Marcos, 2016; Slangen et al., 2014). Thus, the accurate estimation of VLM is vital, not only to disentangle climatic and geodynamic SLC signatures, but also to obtain more robust estimates of past and future relative SLC and its associated uncertainty (Church et al., 2013; Santamaría-Gómez

et al., 2017). In this work, we present a novel approach of VLM estimation using coastal satellite altimetry, tide gauges and the Global Navigation Satellite System (GNSS).

VLM is caused by the superimposition of natural processes and anthropogenic influences in the Earth System and operate on manifold spatial and temporal scales (Pugh and Woodworth, 2014). Mechanisms such as the Glacial Isostatic Adjustment (GIA), the postglacial rebound of the Earth to changing ice and water load, cause distinct large-scale VLM, which can be

assumed to be uniform on centennial timescales (Peltier, 2004). Recent acceleration of land ice mass loss was shown to additionally enhance deformation rates, posing new challenges for sea-level studies due to its time-varying signal (Riva et al., 2017). Surface mass changes are also caused by terrestrial freshwater storage changes and can have small-scale effects on VLM. Groundwater pumping for instance, contributes not only to local small-scale VLM and gravity changes, but also modifies sea level rise in distant areas (e.g., Wada et al., 2012; Veit and Conrad, 2016). Other small-scale VLM effects such as erosion or

tectonic movements can be locally confined to several kilometers with more subtle, not necessarily linear temporal behaviour (Brooks et al., 2007; Kolker et al., 2011; Poitevin et al., 2019). These and other non-linear processes at very short time scales are particularly challenging in the estimation of a linear rate of long-term VLM from TGs, since they would appear as instabilities in the record (similarly to a change in datum).

In response to the substantial impact on relative sea level and the large spectrum of VLM sources, several strategies have

been developed to estimate VLM. The ability to capture the diversity of VLM processes, however, strongly depends on the method and geodetic technique used in the VLM estimation. Furthermore, the coverage and associated accuracy of VLM estimates differ across the methods. Given that the global absolute sea level trend during the altimeter era is in the order of 3 mm/year (3.1 ±0.1 mm/year from 1995 to 2018 as reported in Cazenave et al., 2018), one prerequisite for VLM estimation is that the associated trend uncertainty should be at least one order of magnitude lower than this subtle signal (Wöppelmann

and Marcos, 2016). Hence, dense and accurate VLM estimates are required to complement modelled or measured rates of absolute sea level rates, which is ultimately crucial for coastal planning. Improving the reliability of VLM estimates and their associated uncertainties is thus one major concern of this study. In the following, we briefly contrast the three major approaches of deriving coastal VLM globally.

## 1.1   Estimating Coastal Vertical Land Motions

The majority of global sea-level-studies utilized geodynamic GIA-models to correct, for example, tide gauge records for secular land motion trends or to extrapolate future relative SLC based on climate-projections (e.g., Church and White, 2011; Hay et al., 1990; Carson et al., 2016). GIA still represents the only long-term geological process for which VLM can be modeled on a global scale. However, one caveat is that GIA-VLM models were shown to be still biased by imperfect assumptions of ice history and the Earth's structure (King et al., 2012) and are thus model-dependent (Jevrejeva et al., 2014). Another foreseeable

disadvantage is that the sole application of GIA-models neglects other sources of VLM (e.g. tectonics, erosion or anthropogenic impacts, Wöppelmann and Marcos (2016)). This led, for instance, to discrepancies in estimated rates of historical global mean sea level (GMSL) change, when comparing model-based solutions against those using measurements from GNSS (Hamlington et al., 2016).

For more than a decade, these direct geodetic estimates (GNSS, such as GPS, GLONASS or GALILEO) have been exploited
to determine vertical velocities (Wöppelmann et al., 2007; Snay et al., 2007; Mazzotti et al., 2008). GNSS measurements denote
the most precise source of VLM detection and are well established in local to global scale studies (e.g., Bouin and Wöppel-
mann, 2010; Fenoglio et al., 2012; Santamaría-Gómez et al., 2017). Wöppelmann and Marcos (2016) identified comparably
low formal errors of GNSS-VLM rates (0.21 mm/year) when auto-correlation was taken into account. Santamaría-Gómez et al.
(2012) estimated an accuracy of 0.6 mm/year of GNSS-based VLM (from at least three years of continuous data), by com-
paring 36 globally distributed co-located GNSS velocity estimates. Thus, because of its considerable accuracy, vertical GNSS
velocities frequently served as benchmark estimates for many sea-level applications, e.g. for GIA-model evaluation or local
VLM-corrections of TG records (Sánchez L., 2009; Sanli and Blewitt, 2001).

For the latter use a necessary working hypothesis is that GNSS-VLM represents the same movements as experienced at the
tide gauge (Wöppelmann and Marcos, 2016). Because VLM is shown to potentially possess high spatial variability even on
small scales (tens of kilometers), GNSS-stations should be ideally very close to the tide gauge. This requirement, however,
reduces the number of available co-located stations (130 GNSS stations within a 1 km range of GLOSS (Global Sea Level
Observing System) tide gauges, Wöppelmann et al. (2019)) and thus confine the global coastal coverage to mostly Europe,
Japan and North America.

To extend the number of VLM estimates, several studies advanced the application of combining satellite altimetry (SAT) and
tide gauge (TG) observations (Cazenave et al., 1999; Nerem and Mitchum, 2003; Kuo et al., 2004; Pfeffer and Allemand, 2016;
Wöppelmann and Marcos, 2016; Kleinherenbrink et al., 2018). The principle of this approach is to subtract the absolute SLC
gathered by the altimeter from relative SLC observations at the TG. Ideally, the differenced time series (SAT minus TG) yields
the vertical displacement of the TG with respect to the reference of the altimeter. The accuracy of this technique is, among
numerous other factors, strongly dependent on the stability of the altimeter measurement system. Major systematic errors
stem from limitations in the realization of the reference frame, as well as from limitations in the long-term stability of altimeter
instruments and corrections (e.g., Couhert et al., 2015; Wöppelmann and Marcos, 2016; Watson et al., 2015). Altimetry records
can be affected by globally and regionally varying drifts, which were found to be most pronounced on TOPEX altimeter side-A.
The calibration of altimetry with TGs is influenced by the applied VLM correction (Watson et al., 2015). Conversely, SAT-TG
VLM estimates are affected by the altimeter drift or by errors originating from the intermission drift biases (or drifts w.r.t
the reference mission). Due to the availability of global and continuous absolute sea level measurements, this method not
only provides a complementary source to GNSS measurements, but also improves the geographical distribution of the data, as
virtually every valid TG is usable.

While all of these three sources of information, GIA-models, GNSS and 'satellite altimetry minus tide gauge' (SAT-TG)
techniques have individual merits, their combined use is valuable to further substantiate VLM estimates. GNSS-observations
are necessary to validate both GIA-models and the SAT-TG approach (Santamaría-Gómez et al., 2012; Wöppelmann and
Marcos, 2016; Kleinherenbrink et al., 2018). Recent studies combined all three approaches to reconstruct GMSL (Dangendorf
et al., 2017), or to densify the estimation of contemporary rates of vertical land motions (Pfeffer et al., 2017) or relative sea

level change (Hawkins et al., 2019). Any advancement in these individual approaches, therefore, supports developments of the others and improves the global assessment of coastal VLM estimates.

In this study, we focus on enhancing the application of SAT-TG difference for VLM detection. Our investigations not only further develop the latest progress of the method, but also gain a new perspective on sea level trend and uncertainty characterization and quantification in coastal zones. The next section recapitulates the latest state of $VLM_{SAT-TG}$ estimation on which we base our innovations.

## 1.2 Progress in VLM estimation by satellite altimetry and tide gauge difference

The combination of SAT and TG observations for VLM determination was steadily improved over the last two decades and is elaborated in the latest review by Wöppelmann and Marcos (2016), hereinafter WM16. WM16 investigated performances of different gridded and along-track altimetry products (e.g. AVISO (Archiving, Validation, and Interpretation of Satellite Oceanographic data) and GSFC (Goddard Space Flight Center). They combined sea-level anomalies (SLAs) as 1°-radius averages with monthly-mean TG records from PSMSL (Permanent Service for Mean Sea Level). Among all datasets, the gridded AVISO product revealed the best correlations and residuals between altimetry and TG records. Using this dataset, WM16 also obtained the most precise VLM estimates were achieved, with median formal uncertainties of 0.8 mm/year. Validation against GNSS-based trends from ULR5 (Université de La Rochelle, Institut Géographique National analysis) at 113 colocated stations resulted in an $RMS_{\Delta VLM}$ of 1.47 mm/year of $VLM_{SAT-TG}$ and $VLM_{GNSS}$ trend differences, providing the highest accuracies among the datasets.

Notwithstanding the weaker performance of the along-track product (from GSFC) achieved in WM16, Kleinherenbrink et al. (2018) made great progress in using along-track altimetry (Topex, Jason1 and 2, from the Radar Altimeter Database, RADS (Scharroo et al., 2012)) to estimate VLM. Their approach aimed to overcome the spatial downsampling and associated loss of information in gridded products such as AVISO. They also advanced the procedure of combining altimetry and TG data. Instead of taking 1°-averages around the TGs, they selected altimetry data according to different absolute correlation thresholds and implemented correlation-weighting of time series. Generally, the thresholding strategy functioned as a filter to remove stations of low comparability: With varying correlations between 0.0 and 0.7, they obtained $RMS_{\Delta VLM}$ errors from 2.1 to 1.20 mm/year (at 155 stations), which significantly improved WM16's results. For a consistent set of stations, they found a slight sensitivity of the $RMS_{\Delta VLM}$ to variations of the prescribed minimum correlation, however, they reported insignificant improvements of a few percent. Because they derived GNSS vertical velocities from the Nevada Geodetic Laboratory (NGL) database by taking the median of available estimates within a 50 km range to the TG, they increased the number at which $VLM_{SAT-TG}$-trends could be validated to 155.

Based on Kleinherenbrink et al. (2018) and WM16, we identify two essential factors which are vital for the quality of trend estimation by SAT-TG difference. Advancements with respect to both factors not only potentially led to improved VLM estimates in Kleinherenbrink et al. (2018), but also motivate for further innovations:

1. **Data quality**: In coastal regions accuracy of altimetry measurements is affected by the local departure of the radar signal from the known ocean response (due to inhomogeneities of the illuminated area) and by the inaccuracy of the standard routinely applied corrections and tidal models. Developments for the solution of both issues led to rapid improvements in the recent years by e.g. application of coastal-retracking and advanced geophysical corrections (e.g., Cipollini et al., 2017; Passaro et al., 2014; Fernandes et al., 2015). Dedicated coastal altimetry datasets (e.g. COASTALT, ALES, PIS-TACH) might thus outperform previously applied products (e.g. AVISO), which do not yet benefit from these implementations.

2. **Data selection**: Next to issues concerning data quality, the second factor defining trend uncertainty is the sensitivity of $VLM_{SAT-TG}$ estimates to the spatial selection of altimeter data in the vicinity of the TG. WM16 showed, that averaging SLA in a radius of $1°$ around the TG resulted in higher correlations than using the best correlated or the closest grid point to the TG. Kleinherenbrink et al. (2018) found a small influence of variations of absolute correlation thresholds on the trend estimates. Therefore, considering the diversity of processes which drive coastal sea level variability, such as waves, winds or coastal and bathymetric properties, an advanced adaptation of the choice of altimetry SLA might improve representation of the signal captured by the TG.

These reasons motivate for further improvements of both components, quality of the data and practise of combining altimetry and TG data. We aim to understand how dedicated along-track coastal altimetry can outperform standard-gridded products. We also seek to generalize an optimal selection of SLA's, underpinned by the local dynamical features of measured sea level variability.

In this work, we present a new approach of combining SAT and TG observations to improve VLM estimates. In contrast to previous attempts, we exploit TG and SAT data at the highest available temporal and spatial scale for globally distributed stations. We couple advanced coastal altimetry data with high frequency TG records from the Global Extreme Sea Level Analysis (GESLA). Implementation of these high frequency TG records constitutes a further innovation for VLM estimation. So far such data has only been applied in local studies (Idžanović et al., 2019) and monthly TG data were commonly exploited in this regard. We show that precision and accuracy of the trend estimates can be optimized, when using refined spatial selection criteria of altimetry sea level anomalies. With this approach we identify coherent zones of sea level variability, which best represent the coastal in situ measurements. Our method is generally transferable to analysis of coastal sea level trend determination.

Sections 2 and 3 describe the individual datasets, applied processing steps and the optimization of combining altimetry and TG data. Section 4 presents performance of trend estimates, i.e. estimated uncertainties and validation against GNSS data (in this study all GNSS data are based on the Global Positioning System (GPS)). Finally, we contrast our results and methods with previous work and discuss the impact of the interconnection of time and space-scales on the evolution of coherency of sea level in coastal regions (section 5).

## 2  Data

We use different altimetry products, in order to assess the impact of special coastal products on associated $VLM_{SAT-TG}$ trend estimates. We compare the coast-dedicated retracker ALES (Adaptive Leading Edge Subwaveform retracker, Passaro et al. (2014)) along-track product against the interpolated AVISO dataset (sections 2.1 and 2.2). Altimetry data are combined with

TG observations from the monthly mean PSMSL and the high frequency GESLA data base, which are described in sections 2.3 and 2.4. We develop a new coupling strategy of high-rate altimetry and TG records in section 3.2.

### 2.1  Coastal along-track altimetry - ALES

The coastal altimetry product is constructed from 1-Hz multi-mission altimetry measurements processed by DGFI-TUM (OpenADB, https://openadb.dgfi.tum.de). We combine data from the missions ERS-2, Envisat, Saral, Jason1-Jason3 and their

extended missions, which provide continuous altimetry time series of 23 years (1995-2018). For all missions, satellite orbits in ITRF2008 are used, mostly processed by CNES (GDR-E). For ERS-2, GFZ VER11 orbits are applied. The SGDR data are re-processed with the ALES retracker (Passaro et al., 2014) and an improved sea state bias correction scheme (Passaro et al., 2018). The geophysical corrections are summarized in Table 1. They are consistent with those incorporated in the latest development of the empirical ocean tide model (EOT19p) by Piccioni et al. (2019). If available, the Dynamic Atmopsheric

Correction consists of the ECMWF ERA-Interim reanalysis (DAC-ERA, Carrère et al. (2016)). This product especially reduces along-track sea level errors in the earlier missions (in this study ERS-2). Because this product is unavailable for the very recent missions, we implement the DAC (Carrère and Lyard, 2003) based on ECMWF for the last cycles of Jason-2 (and its extended mission) and the full Jason-3 and Saral missions. To reduce radial errors in the different missions, the tailored coastal altimetry product is cross-calibrated using the global multi-mission crossover analysis (MMXO) (Bosch and Savcenko, 2007;

Bosch et al., 2014). The MMXO minimizes a large set of globally distributed single- and dual sea surface height crossover differences by least-squares adjustment. The estimated radial errors are used to correct each individual sea surface height measurement. In this way, we not only reduce orbit inconsistencies, but also those originating from the range and from applied corrections. Since we estimate a radial correction for each observation, we minimize intermission drift differences as well as regionally correlated errors. Note that this approach is a relative calibration and provides range bias corrections with respect to

NASA/CNES reference missions. Any remaining absolute drift of these reference missions (with respect to TGs) still influence the drift of the whole altimeter solution.

We map all altimetry records on 1-Hz nominal tracks consistent with the CTOH nominal paths (Center for Topographic studies of the Ocean and Hydrosphere, www.ctoh.legos.obs-mip.fr) of the individual missions, using nearest-neighbour interpolation. Then, we scan the data for outliers along the tracks, to hinder spurious extreme values to propagate in time series.

This scheme features:

– Absolute thresholds: Any absolute SLA exceeding 2 m is excluded.

**Table 1.** Applied models and geophysical corrections for estimating sea level anomalies.

| Parameter | Model | reference |
| --- | --- | --- |
| Range and Sea State Bias | ALES | (Passaro et al., 2014) |
| Inverse barometer | DAC-ERA | (Carrère et al., 2016) |
| Wet troposphere | GPD+ | (Fernandes et al., 2015) |
| Dry troposphere | VMF3 | (Landskron and Böhm, 2018) |
| Ionosphere | NIC09 | (Scharroo and Smith, 2010) |
| Ocean and Load tide | FES2014 | (Carrère et al., 2015) |
| Solid Earth and Pole tide | IERS 2010 | (Petit and Luzum, 2010) |
| Mean Sea surface | DTU18MSS | (Andersen et al., 2018) |

– Running median test: If the absolute difference of the data and its running median (centered, over 20 points) is greater than 12 cm, data are excluded.

– Consecutive difference test: Outliers are detected when the difference of consecutive points exceeds 8 cm. The test identifies the outliers according to the differences of the other neighbouring values

The absolute thresholds (12 cm, 8 cm) correspond to 2-$\sigma$ of the median running variability and 2-$\sigma$ of absolute consecutive differences based on the analysis of different tracks of Jason-2 and ERS-2.

SLAs along the same track and cycle are then averaged over predefined areas as described in sections 2.5 and 2.6. We built a time series by considering all averaged SLAs from the along-track multi-mission dataset for the study period. To check for outliers in each SLA time series, we exclude values exceeding absolute values of 3-$\sigma$ of the data. This cleaned 1 Hz coastal altimeter dataset is hereinafter called ALES and used for the combination with the TG-datasets described in section 3.1.

## 2.2 Gridded altimetry data - AVISO

The gridded Ssalto/Duacs altimeter product was produced and distributed by the Copernicus Marine Environment Monitoring Service (CMEMS, http://marine.copernicus.eu) and is hereinafter called AVISO as it was previously distributed by CNES AVISO+. We use monthly sea level anomalies, which are resolved on a 0.25° Cartesian grid and cover the period from 1992-2019. The product already includes the DAC (Carrère and Lyard, 2003) comprising the dynamical barotropic ocean response to atmospheric forcing (modelled with MOG2D-G), as well as the inverse barometer (IB) response. Consistent with the along-track dataset ALES, FES2014 (Carrère et al., 2015) is implemented to correct for tidal signals. Other corrections and pre-processing steps are documented by CMEMS.

## 2.3 Monthly tide gauge data - PSMSL

We use monthly mean TG data from the datum controlled PSMSL (Holgate et al., 2013) database. The PSMSL constitutes the primary source of TG data for most sea-level research, or for the assessment of long-term trends of VLM based on SAT and

TGs. The service undertakes quality control of the data including checks for consistency of the annual cycle, outlier detection or intercomparisons with neighbouring stations, which enhances the reliability of the data. Among all available stations, we

select those which contain at least 180 months (15 years) of valid measurements during the altimetric era (1993 - present), resulting in a total number of 627 stations. We apply the same monthly-averaged DAC-correction as used for the AVISO data (Carrère and Lyard, 2003). To match the DAC-correction with the TG records, we select among the 9-closest grid-points of the solution, the one which results in the highest variance reduction.

## 2.4    High frequency tide gauge data - GESLA

In addition to monthly-mean PSMSL TG-data, we exploit the GESLA dataset (Woodworth et al., 2016), which contains a large global collection of high-frequency TG records with sampling rates ranging from hours down to 6 minutes. The latest version GESLA 2 contains in total 1355 station records and was assembled from a variety of international and national databanks (e.g. UHSLC (University of Hawaii Sea Level Center) and GLOSS) or independent sources. It thus also shares many stations with the monthly PSMSL database, the preferred dataset for $VLM_{SAT-TG}$ computation. Unfortunately, at this time, GESLA holds

only data until 2015. Therefore, we also restrict the considered period for all dataset combinations (see section 3.1) to before 2015. As for the PSMSL data, we select stations with at least 180 months of valid data.

In contrast to PSMSL data used in WM16, GESLA TGs feature no rigorous outlier rejection by default in except that of the primary data providers (Woodworth et al., 2016). Extreme values from strong signals like tsunamis or station shifts and other irregularities are still present in the data. Some of those issues are adressed on the GESLA-webpage, however, for the sake

of long-term trend evaluation, we perform a further global outlier analysis. Therefore, we check all TG time series manually for irregularities: Station shifts from seasonal to interannual timescales are either handled by dismissing certain sections of the time series or completely excluding the TG from the analysis. Single extreme events from hourly to monthly timescales are only excluded, when they deviate from the measurements by several meters, because we want to maintain as much data as possible. If such events are present, we flag any values beyond the upper/lower 0.999 quantiles of a fitted normal distribution

of the data. Occasionally we apply this quantile-outlier exclusion recursively.

To obtain a uniform temporal resolution, we resample this outlier-free TG set to hourly records by cubic interpolation. The records are then corrected for the tidal signal as well as for the ocean response to atmospheric wind and pressure forcing. The tidal variability is suppressed by using a 40-h loess (locally estimated scatterplot smoothing) filter (Cleveland and Devlin, 1988) as in Saraceno et al. (2008). This filtering approach most effectively reduces tidal variance at periods lower than  2

days (e.g. reduction by more than two orders of magnitudes at daily periods). However, tidal variability at periods larger than 2 days is not significantly attenuated by the filter. Therefore, one caveat of this approach is that there remains residual tidal variance at longer periods between TGs and altimetry, given that the latter features a model-based adjustment for longer tides. We do, however, not apply the same tidal model to the TGs, due to known issues related to decreased model performance in shallow water (Piccioni et al., 2018). In accordance with PSMSL-TG data, we implement the same Dynamic Atmospheric

Correction (Carrère and Lyard, 2003). This solution features a 6h sampling frequency, which is therefore down-sampled to hourly anomalies by cubic interpolation. For the global dataset, we obtain a mean variance reduction of 37.8% and a mean

correlation of 0.6. As in WM16 and Ponte (2006), we find a distinct latitude dependence of correlations and variance reduction, with decreasing performance nearer to the equator. We note, that the total variance reduction, which we apply on the high-rate TG data is naturally less than in WM16, who corrected monthly mean, detrended and deseasoned data.

 ## 3  Methods

### 3.1  Dataset combinations

To understand the sensitivity of the VLM estimations on (1) quality and resolution of the data and (2) the selection procedure, we analyse the performances of four different dataset combinations: ALES-PSMSL-250km, ALES-GESLA-250km, AVISO-PSMSL-250km and ALES-GESLA-ZOI.

The first three combinations are constructed to compare the performances of the along-track (ALES) against the gridded altimetry product (AVISO) combined with monthly TG observations. With ALES-GESLA-250km we also investigate the possible advantage of using the GESLA high rate TG product. For all these experimental sets, SLA time series are merged or averaged within a 250 km radius around the TGs, which is thus a selection procedure independent of the actual comparability of SLAs.

To produce the ALES-GESLA-250km dataset, we derive differences of the merged, non-uniformly sampled SLAs and the hourly-sampled GESLA TG records, by cubic interpolation of the latter and a maximum allowed time-lag of 3 hours between the measurements. We down-sample these high-rate differenced time series to monthly means. For ALES-PSMSL-250km on the other hand, we first compute monthly-means from SLAs and subsequently subtract these monthly SLAs from the monthly-sampled relative SLAs from PSMSL. Finally, we directly compute the differenced SAT-TG time series from the averaged
monthly AVISO and the PSMSL data, which yields the AVISO-PSMSL-250km dataset.

Using these combinations, we investigate the mere changes from differences formed using along-track data at high or at low frequency (ALES-GESLA-250km and ALES-PSMSL-250km), or using monthly gridded data (AVISO-PSMSL-250km). Here, 'high-frequency' refers to daily time scales of variability and 'low-frequency' to monthly time scales. The dataset ALES-GESLA-ZOI incorporates further SLA-selection-schemes, which are explained in the following section.

### 3.2  The Zone of Influence

We aim to develop a new SLA-selection scheme, which accounts for the observed coherency of sea level variability. However, due to the diversity of the underlying physical mechanisms and their complex interplay with the coast, the spatial coherency of sea level dynamics is highly variable in coastal regions (Woodworth et al., 2019). Coastally trapped waves, for instance, were argued to establish long-range correlations along the continental slopes (Hughes and Meredith, 2006) and to mediate
the influence of the open ocean (Hughes et al., 2019) on the coast. While some signals, such as interannual modes of climate variability, generate high spatial coherence, other local features, such as the presence of a coastal current, can significantly modify the sea level variability within few kms of the coast, as shown in the case of the seasonal signal of the Norwegian

Coastal Current in Passaro et al. (2015). Accordingly, the capability of compare TG-based sea level variability with altimetry utterly depends on which time and length scales are resolved by the data.

The key concept of our approach is to capture the extent to which coastal altimetry measurements are similar to the in situ TG observations. To do so, we extend the methodology proposed by Santamaría-Gómez et al. (2014), who looked for the altimetry grid point mostly correlated with the TG, and Kleinherenbrink et al. (2018), who considered a larger set of points based on absolute thresholds of correlation. In contrast to these previous studies, we assess the influence of using relative thresholds of comparability on both the accuracy and the uncertainty of the trends.

We exploit combinations of along-track ALES data and high frequency GESLA records, to identify regions of sea level variability that show maximum coherency with TG observations, which we hereinafter call the Zone of Influence (ZOI). With this approach, our objective is to decrease noise of the differenced, high frequency $VLM_{SAT-TG}$ time series using the ZOI to hone trends and uncertainty estimates.

To define the ZOI, we investigate different statistical criteria $S$, which provide a measure of similarity of sea level variability

between TG and SAT observations. Here, we use the Pearson correlation coefficient, the $RMS_{SAT-TG}$ as well as the amplitude of the residual annual cycle between both TG and SAT records. We compute each of those measures for every point of the 1 Hz along-track data (ALES) in combination with the TG records from GESLA. As for ALES-GESLA-250km, TG data are interpolated onto the time step of the altimetry records. Correlations and $RMS_{SAT-TG}$ are computed from the detrended TG and SAT time series. The amplitude of the residual annual cycle is obtained from the remaining seasonal signal of the difference of the time series (SAT-TG). We acquire a dataset, holding information of the performance of multi-mission along-track data

in the vicinity of every GESLA TG. The statistics are based on de-trended data. Thus, all the metrics may be influenced by the similarity of the annual cycle. However, by repeating this analysis using de-trended and de-seasoned data (not shown), no significant differences are identified.

To confine the ZOI, we select sub-sets of the data containing the best-performing statistics (i.e. highest correlation, lowest

$RMS_{SAT-TG}$ or residual annual cycle) above the $Xth$-percentile according to the distribution of the statistic $S$ in a 300 km radius around the TGs. Every sub-set $(X, S)$ represents an individual ZOI, in which we average SLAs in accordance with the steps involved in the aforementioned 250 km-radius-selection (ALES-GESLA-250km, section 3.1). The high-rate SLA time series (ALES) are then again subtracted from GESLA, providing the ALES-GESLA-ZOI dataset for VLM estimation (section 3.3).

Note that in contrast to the 250 km selection, we extend the range in which SLAs are taken into account to 300 km to define

the ZOI. The previous 250 km selection is, as in Kleinherenbrink et al. (2018), based on the space auto-correlation scales of SLAs, which reflect characteristic eddy length scales (Stammer and Böning, 1992; Ducet et al., 2000). These scales decrease towards higher latitudes (i.e. towards the poles) with changing internal Rossby radius. However, several studies found much larger correlation length scales of SLAs, in particular along shorelines (Calafat et al., 2018; Hughes and Meredith, 2006). Other mechanisms than mesoscale eddy activity were investigated to account for these coherent changes. One example is given

by Calafat et al. (2018), who analysed the driving factors of sea-level variability at the south-eastern coast of the US. Using altimetry and three different ocean-models, they found coherent changes of the annual amplitude of SLAs over length scales of thousands of kilometers along the coast from the Yucatan Peninsula to Cape Hatteras. While the annual cycle signal itself was

dominated by steric changes, with likewise large-scale correlations at the continental slope, changes of the annual amplitude were argued to be dominated by boundary waves exerted by incident Rossby waves. Because we similarly find correlations

beyond the 250 km length-scale, in particular along elongated coastal regions (Figure 1 a and b), we justify the larger 300 km radius.

We identify coherent zones of sea-level variability represented by different selection-criteria in Figure 1. The statistics $S$ are computed based on individual along-track SLA time series (ALES) and GESLA TGs. We show different maps of these along-track statistics for (a) the Australian Coast, (b) Californian Coast and (c) Chichijima island (Japan). The contour in the

first column exemplifies the extend of a ZOIs, which represents a sub-set of 20% of the best correlated data.

The obtained coherent structures reveal notable dependencies on the local bathymetric and coastal properties. Figure 1 (a), for instance, shows far-reaching alongshore correlations, which is supported by all of the analysed selection criteria. In this example, the separation of the region of the coastal shelf-sea dynamics from region of offshore variability is in good agreement with the underlying bathymetric gradients. Kurapov et al. (2016) found similarly pronounced SLA coherency along for the

Californian coast, as shown in Figure 1b). Based on model data and TG observations, they explained the large-scale along-shore correlation pattern in part with the propagation of coastal trapped waves. In other locations such as in Chichijima island (see Figure 1c), coastal and bathymetric control of SL is reduced and different structures of coherency evolve. Consequently, the ZOI can strongly vary in shape depending on the local coastal features and drivers of coastal variability.

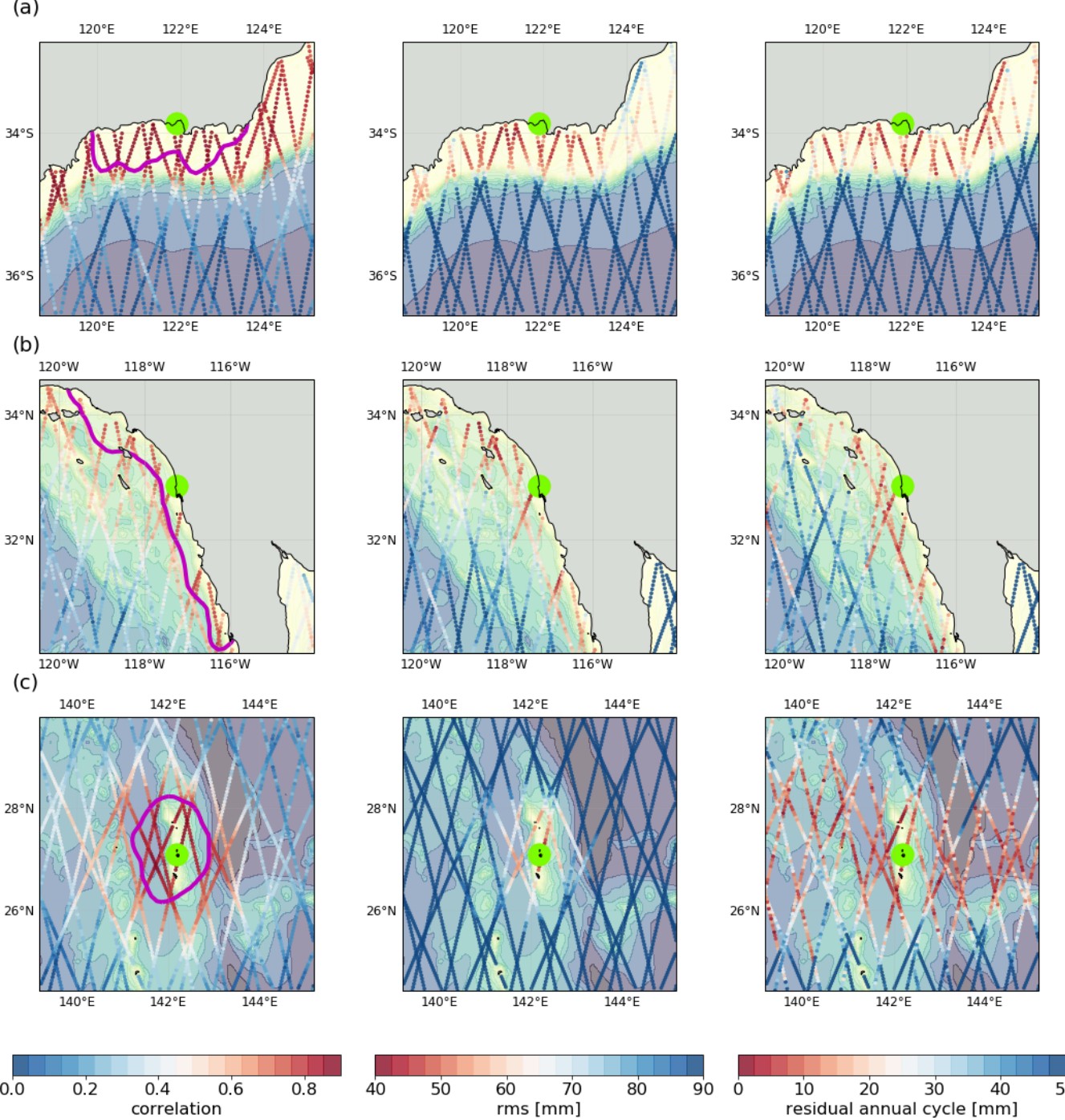

**Figure 1.** Zone of Influence: Different coherent zones of sea level variability are identified by different statistical criteria $S$. The columns show correlations, $\mathrm{RMS_{SAT\text{-}TG}}$ and the residual annual cycle from left to right. The metrics are computed on every point of the 1 Hz along-track product, comparing the performance of altimetry measurements with the TGs, highlighted in green (center). (a) shows the South-Coast of Western Australia, (b) the western coast of North America (TG in San Diego) and (c) Chichijima island (Japan). The 'color' contour in the first column indicates a Zone of Influence built from 20% of the best-correlated SLAs within a 300 km radius. The underlying contours denote the underlying bathymetry.

Comparing these three examples, we also observe that absolute values of the statistics differ from site to site. Correlations of along-track data near the Australian coastline, for instance, outperform the ones in example Figure 1b. The same holds for RMS$_{\text{SAT-TG}}$ values. These differences not only indicate different degrees of coherency, but can also stem from regional deviations in the quality of data, i.e. quality of TG records or error sources in the altimetric product, such as tidal adjustments or coastal corrections. Differences can also be caused by coastal properties, e.g. when the TGs are located in sheltered areas, which separates the in situ variability from the one measured at distant altimeter tracks. We analyse therefore the use of relative thresholds, to select the SLAs, since setting absolute thresholds as in Kleinherenbrink et al. (2018) might not be applicable in all cases. Figure 1c) also shows that different statistics can determine different extents of the ZOI, considering that rather poorly correlated areas are partially characterised by low residual annual cycle amplitudes.

A correct choice of the ZOI based on a sub-set of high performance SLAs can significantly reduce the SAT-TG residuals as are exemplified in Figure 2. Here, we show three time series of SAT-TG differences for the Australian site (see Figure 1a). The first series (Figure 2a) indicates much lower residual noise, when the time series is constructed from the 20% best SLAs (according to the RMS$_{\text{SAT-TG}}$). Here, the ALES-GESLA-ZOI residuals outperform those of the other combinations ALES-PSMSL-250km and AVISO-PSMSL-250km, which are still affected by a pronounced annual cycle not related to VLM.

While using relative thresholds can reduce the noise of VLM$_{\text{SAT-TG}}$ time series for individual stations, we seek to identify a globally optimal ZOI definition and associated criteria and thresholds, which lead to largest improvements of uncertainty and accuracy of VLM$_{\text{SAT-TG}}$. Therefore, we vary the relative thresholds $X$ between 0.0 and 0.975 (with a stepsize of 0.025), which refers to using 100% and 2.5% of the best performing SLAs according to each criteria. For each threshold and criterion we derive an individual global VLM$_{\text{SAT-TG}}$ trend and uncertainty dataset. We validate the performance of the trend estimates for a specific ZOI definition in accordance with section 3.4. Optimal parameter $X, S$ are then suggested for the global application (section 4.2).

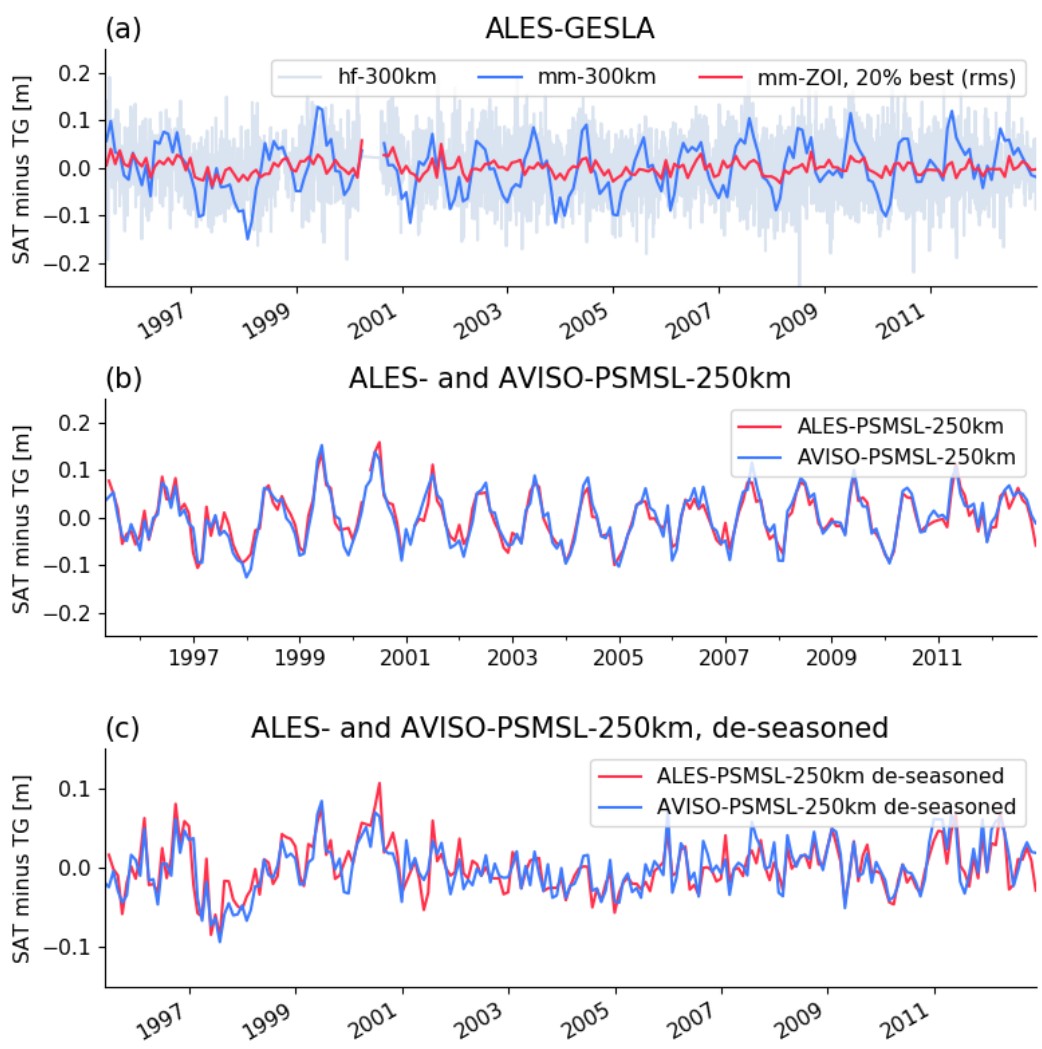

**Figure 2.** Shown are 'SAT minus TG' time series for different datasets and configurations for the TG in Figure 1a). (a) Monthly mean (mm) time series for ALES-GESLA, when all SLAs are averaged in a 300 km radius (blue) and when SLAs are comprised of the 20% most representative anomalies based on the RMS$_{\text{SAT-TG}}$ between altimetry and TG (red). The grey line denotes the underlying high-frequency time series. (b) Monthly mean differenced time series for ALES-PSMSL and AVISO-PSMSL, which are based on a 250km-radius selection of SLAs. (c) Same as (b) but with the annual and semi-annual signals removed.

 ### 3.3 Statistical analysis: Trend and uncertainty estimation

We fit the differenced time series to a combination of a deterministic model and stochastic noise models with the Maximum Likelihood Estimation (MLE) method. Parameters of the deterministic model are comprised of a constant offset $A$ and a linear trend $B$. The annual and semi-annual signals are expressed by harmonic functions with the annual and semi-annual frequencies $\omega_{1,2}$ and amplitudes $C_{1,2}$ and $D_{1,2}$.

$$y(t) = A + Bt + \sum_{i=1}^{2} C_i \cos 2\pi t \omega_i + D_i \sin 2\pi t \omega_i \qquad (1)$$

When combining altimetry and TGs for VLM estimation, several sources can contaminate the differenced time series and inflate the actual 'red'-noise (low-frequency) content in the residuals, which generates auto-correlated signals in the data. The SLA computation is affected by the instrumental errors of the range estimation and of each of the geophysical corrections (Ablain et al., 2009). Such errors, as well as the measurement error of the TG itself, show up as residuals in the differenced time series. Moreover, sea level dynamics that are not common between the TG and altimeter observation locations will also contribute to the SAT-TG differences. Therefore, to avoid underestimation of the uncertainty of the parameters, we take into account auto-correlation in the residuals of the detrended and deseasoned time series. We describe the power spectral density of the noise with a combination of a power-law and a white noise model (using the Hector software (Bos and Fernandes, 2019)). The power-law process assumes that time-correlated noise power is proportional to $f^{\kappa}$, which for negative spectral indices $\kappa$ describes increasing power at lower frequencies $f$ and a white-noise process when $\kappa = 0$ (Agnew, 1992). Santamaría-Gómez et al. (2011) showed that this combination (of power-law and white noise model) represents the best approximation of the noise content for 275 GNSS station position time series. This combination was also implemented in studies concerned with $\text{VLM}_{\text{SAT-TG}}$ estimation (WM16, Kleinherenbrink et al. (2018); Ballu et al. (2019)). In particular, the spectral index $\kappa$ can contribute to detect the intrusion of low-frequency signals in the differenced time series. Next to the spectral index $\kappa$ we estimate the individual fractions of the power-law and white noise models, as well as the total variance $\sigma^2$ which scales the amplitude of the noise. We emphasize that for individual regions other noise models could be more appropriate than the implemented PL+WN model and would thus yield more realistic uncertainty estimates. An advanced regional spectral analysis to identify the most suitable models is however beyond the scope of this study.

### 3.4 Validation of $\text{VLM}_{\text{SAT-TG}}$ with $\text{VLM}_{\text{GNSS}}$ trends

To validate SAT-TG-based trend estimates, we use the ULR6a GPS solution provided by the GNSS data assembly centre SONEL (Systeme d'Observation du Niveau des Eaux Littorales, http://www.sonel.org). The reanalysis covers 19 years of GNSS data from 1995 to 2014, which are processed within the ITRF2008, consistent with the reference frame of altimetry orbits. The primary coordinates provided by GNSS are geocentric Cartesian coordinates (X, Y, Z, Vx, Vy, Vz). For the comparison with vertical trends inferred from other techniques, they are converted to ellipsoidal coordinates (latitude $\phi$, longitude $\lambda$ and ellipsoidal height h, and $V_\phi$. $V_\lambda$, $V_h$). Thus, we compare GNSS ellipsoidal height trends ($V_h$) with SAT-TG trends.

It should be mentioned that, while the altimetry trends refer to the so-called TOPEX/Poseidon ellipsoid, the GNSS vertical trends refer to the GRS80 (Geodetic Reference System, 1980; Moritz (2000)) ellipsoid. Although there is difference of 70 cm between the semi-major axes of both ellipsoids, the GNSS and SAT vertical trends can be compared without degradation of precision, as both ellipsoids are geocentric and have the same orientation with respect to the Earth's body (e.g., the ellipsoid minor axes coincide with the mean Earth's rotation axis, and the major axes are on the Earth's equatorial plane). We take into account GNSS stations which are closer than 1 km to a TG. With this constraint we aim to avoid potential differential vertical motions between the TG and the GNSS-antenna (WM16).

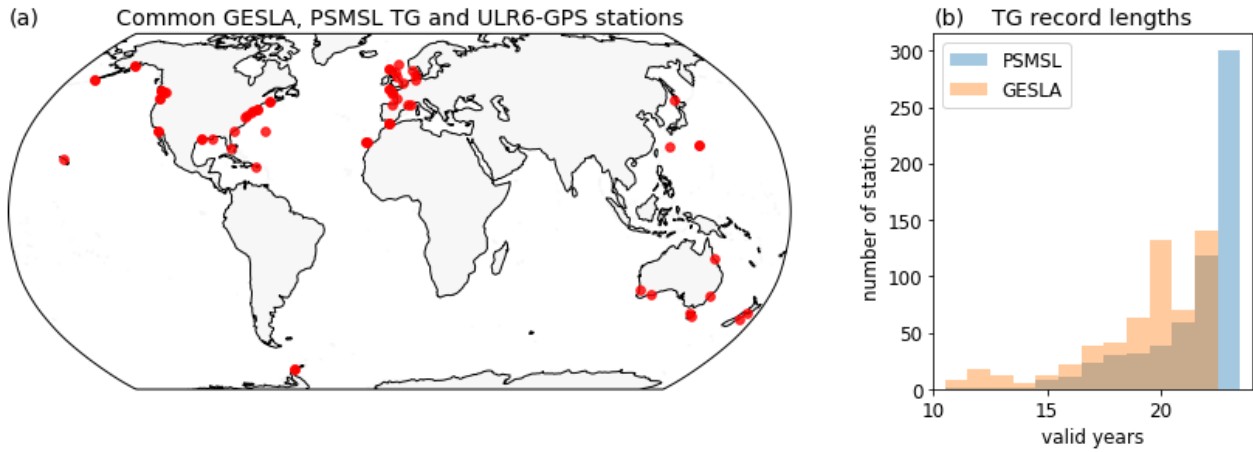

**Figure 3.** a) Global Distribution of 52 common GESLA, PSMSL and ULR6-GNSS stations, which meet the described requirements. b) Number of all TG stations sorted by the amount of months which contain valid data (here shown in valid years) in the period 1993-2015.

The TG locations and record lengths differ among the presented experimental datasets (section 3.1). Therefore, we define several requirements for the validation of those experimental-datasets, to obtain a consistent set of TG and GNSS validation pairs. In contrast to PSMSL records GESLA-TG observations only last until 2015. Even when PSMSL TG records are limited to before 2015, they still contain more months of valid data than GESLA (see Figure 3b). Hence, we align the time period covered by the PSMSL-TGs to the corresponding GESLA-TGs for all following experimental datasets. Generally, we only take into account SAT-TG time series, when they cover at least 120 months of valid data. Note, that the outlier analysis (section 2.4) or coupling of high frequency TG data in the ZOI can reduce the length of the SAT-TG time series for GESLA TGs. Taking into account all these requirements, we obtain 52 common GESLA and PSMSL TGs, which provide a neighbouring GNSS station within 1 km distance. These pairs are validated for ALES-PSMSL-250km, ALES-GESLA-250km and AVISO-PSMSL-250km. The ALES-GESLA-ZOI combination includes six more stations. The resulting validation pairs are shown in Figure 3a), where a higher coverage in northerly and midlatitude regions is evident.

We compute the RMS$_{\Delta VLM}$ and the median of the differences ($\Delta$VLM) of VLM$_{SAT\text{-}TG}$ and VLM$_{GNSS}$ for a given dataset

combination. To take into account the derived formal errors ($U$) of the estimate we compute the weighted RMS$_w$ as follows:

$$RMS_\mathrm{w} = \sqrt{\sum_{i=1}^{n}(w_i(VLM_{GNSS_i} - VLM_{SATTG_i})^2)} \qquad (2)$$

with weights

$$w_i = \frac{\sqrt{(U_{GNSS_i}^2 + U_{SATTG_i}^2)}^{-1}}{\sum_{i=1}^{n}(\sqrt{(U_{GNSS_i}^2 + U_{SATTG_i}^2)}^{-1})} \qquad (3)$$

We analyse also the median of the absolute value of differences (|$\Delta$VLM|). This metric is less prone to extreme devia-

tions and can thus consolidate the evaluation of the dataset performances. We generally assume that GNSS provides a more accurate estimation of the linear component of the VLM with a smaller error than VLM$_{SAT\text{-}TG}$, despite shorter time span of measurements. Hence, for the purposes of this paper and as done in all studies concerning VLM$_{SAT\text{-}TG}$ estimation, we define as measures of accuracy the RMS$_{\Delta VLM}$ and additionally the median of |$\Delta$VLM|. We include the spectral index $\kappa$ (see section 3.3) as it helps to understand the level of auto-correlation of the time series. All statistics other than the RMS$_{\Delta VLM}$ denote median

values (of all VLM$_{SAT\text{-}TG}$ estimates) for a specific dataset configuration.

## 4 Results

### 4.1 Comparison of different datasets configurations based on a 250 km average selection

We compare performances of the three datasets which are constructed from 250 km radius SLA averages in Table 2. Validation against GNSS vertical velocities reveals that the gridded combination AVISO-PSMSL-250km slightly outperforms ALES-

PSMSL-250km in terms of accuracy. Both the RMS$_{\Delta VLM}$ and the median of absolute trend differences are 9% lower for AVISO-PSMSL-250km. This confirms that, if all the available altimetry data within a wide region are compared against monthly values of TGs, the use of a gridded product outperforms the along-track performances (WM16). Kleinherenbrink et al. (2018) similarly compared an along-track combination of 250 km-SLA averages (from RADS) and PSMSL TG data with the AVISO-PSMSL combination from WM16. They found a small RMS$_{\Delta VLM}$ reduction of 0.1 mm/year when using the

along-track product without any correlation thresholds applied. WM16's trends were however based on 1$^\circ$ radius-averages of SLAs (in contrast to the 250 km selection), and record lengths were not equalized as in this study.

For both combinations the median of the VLM differences (ALES-PSMSL-250km: -0.87 mm/year; AVISO-PSMSL-250km: 0.56 mm/year) deviates from values shown in previous studies [WM16: -0.25 mm/year and, Kleinherenbrink et al. (2018): -0.06 mm/year]. In contrast to these previous estimates, we use different spatial selection scales of SLAs, smaller numbers of

TG-GNSS pairs and deviating record lengths, which impedes a direct comparison. Moreover, the altimetry datasets might be

affected by instrumental drifts. In this respect, differences among the datasets may be caused not only by different techniques applied to reduce intermission biases (e.g., the MMXO approach for ALES), but also by different missions incorporated in the records. Note that in contrast to ALES, AVISO contains TOPEX, which has also been shown to be affected by a strong drift (Watson et al., 2015). Still, the observed $RMS_{\Delta VLM}$ of AVISO-PSMSL-250km (1.50 mm/year) is comparable to WM16 result

(1.47 mm/year). In contrast to trend accuracies, the uncertainties are 5% lower for ALES-PSMSL-250km than for AVISO-PSMSL-250km. As in WM16, the spectral index $\kappa$ of the interpolated gridded product is lower than for the along-track data. Both $\kappa$ indices (-0.56 and -0.39) also match well those found by WM16 for AVISO (-0.5) and the along-track product (-0.4, GSFC). The larger spectral index (-0.39) is associated with reduced power of the noise at low frequencies and thus indicates reduced contamination of the SLA signal by sea-level variations that do not represent those measured at the TG. This enhanced

comparability is also reflected in the lower trend uncertainties of ALES-PSMSL-250km (0.69 mm/year) compared to AVISO-PSMSL-250km (0.73 mm/year). The differences between the characteristics of the residuals of the datasets can partially be explained by the resolution of the data: Due to the spatial filtering of the data, the gridded solution AVISO incorporates information of SLAs beyond the 250 km radius and thus contains time-correlated SL-signals with stronger deviations from the TG records.

**Table 2.** Statistics of different SAT-TG combinations. $\Delta$VLM refers to the differences of $VLM_{SAT-TG}$ and $VLM_{GNSS}$ trends. $X$ denotes the relative level of comparability above which data is included.

| X | $RMS_{\Delta VLM}$ mm/year | weighted $RMS_{\Delta VLM}$ mm/year | med. $|\Delta VLM|$ mm/year | med. $\Delta VLM$ mm/year | med. uncertainty | spectral index $\kappa$ |
|---|---|---|---|---|---|---|
| \multicolumn{7}{c}{**ALES-PSMSL-250km** (52 stations)} ||||||| 
|  | 1.68 | 1.57 | 1.28 | -0.87 | 0.69 | -0.39 |
| \multicolumn{7}{c}{**AVISO-PSMSL-250km** (52 stations)} ||||||| 
|  | 1.50 | 1.48 | 1.12 | 0.56 | 0.73 | -0.56 |
| \multicolumn{7}{c}{**ALES-GESLA-250km** (52 stations)} ||||||| 
|  | 1.51 | 1.47 | 1.14 | -0.39 | 0.79 | -0.39 |
| \multicolumn{7}{c}{**ALES-GESLA-ZOI** (best $RMS_{SAT-TG}$, 58 stations)} ||||||| 
| 0 | 1.54 | 1.45 | 0.98 | -0.46 | 0.86 | -0.45 |
| 0.1 | 1.39 | 1.36 | 0.9 | -0.27 | 0.86 | -0.44 |
| 0.2 | 1.34 | 1.33 | 0.88 | -0.36 | 0.83 | -0.47 |
| 0.3 | 1.32 | 1.36 | 0.83 | -0.44 | 0.78 | -0.46 |
| 0.4 | 1.3 | 1.38 | 0.87 | -0.37 | 0.76 | -0.45 |
| 0.5 | 1.29 | 1.4 | 0.86 | -0.26 | 0.73 | -0.47 |
| 0.6 | 1.3 | 1.43 | 0.87 | -0.31 | 0.71 | -0.47 |
| 0.7 | 1.28 | 1.39 | 0.82 | -0.41 | 0.66 | -0.48 |
| 0.8 | 1.28 | 1.37 | 0.86 | -0.41 | 0.58 | -0.43 |
| 0.9 | 1.53 | 1.58 | 0.97 | -0.43 | 0.61 | -0.46 |

In comparison with the low-frequency datasets (ALES-PSMSL-250km and AVISO-PSMSL-250km), the high-rate set-up
ALES-GESLA-250km improves the $RMS_{\Delta VLM}$. The absolute bias of trend differences decreases more substantially to 0.39
mm/year (compared to -0.87 mm/year). Compared to ALES-PSMSL-250km, we find increased trend uncertainties for ALES-
GESLA-250km, which can be partially explained by higher power-law variance of this GESLA-based configuration. Although
trend uncertainties are higher for the ALES-GESLA-250km configuration, we choose this set-up to investigate the impact of
the ZOI. This dataset provides better results concerning trend accuracy (weighted or unweighted RMS) and has a lower median
bias. Moreover, using the high-frequency data, we are able to couple SAT and TG observations at much higher temporal reso-
lution than it would be the case when using monthly PSMSL data. Therefore the ALES-GESLA coupling is further developed
based on a better definition of the ZOI in the next section.

### 4.2    The Zone of influence improves VLM estimates

We investigate how the ZOI selection of SLAs fosters quality of SAT-TG VLM estimates. As addressed in section 3.2, we build
the ZOI upon different criteria of comparability: $RMS_{SAT-TG}$, correlation and the residual annual cycle. First, we focus on the
results of using the $RMS_{SAT-TG}$ of the detrended differenced time series (Table 2 and Figure 4, ALES-GESLA-ZOI). We observe
that the $RMS_{\Delta VLM}$, the median of absolute and total differences, as well as trend uncertainties decrease towards higher relative
thresholds. The statistics converge to a minimum when the ZOI is restricted to the 30-20% best data. To compare ALES-
GESLA-ZOI with the other dataset combinations, we compute the statistics for the same 52 TGs used in these configurations
(because the shown statistic in Table 2 refer to a larger set of 58 stations). At the 20% thresholds, we obtain similar performances
with a $RMS_{\Delta VLM}$ of 1.29 mm/year, median uncertainty of 0.51 mm/year and a median of absolute differences ($|\Delta VLM|$) of
0.86 mm/year. Thus, the improvements of $RMS_{\Delta VLM}$ compared to the plain 250 km-radius selection (ALES-GESLA-250km)
is 15% and 35% for uncertainties. Hence, we find more substantial, nearly linear reductions of trend uncertainty with increasing
relative thresholds compared to trend accuracy ($RMS_{\Delta VLM}$, Table 2, ALES-GESLA-ZOI). As demonstrated for different time
series in Figure 2, selecting e.g. highly correlated SLAs efficiently reduces the noise of the residuals. Correspondingly, at higher
levels of comparability, the variance, which scales the amplitudes of the considered noise models, decreases (not shown).

    Because the spectral index (for ALES-GESLA-ZOI) is slightly lower (-0.43 at 20% level) than for ALES-GESLA-250km
(-0.39) it cannot account for the uncertainty improvements. Here, the lower $\kappa$ index reveals an relative increase of power
at low frequency (i.e. time scales longer than months). Thus the bulk of improvements we see in uncertainty (comparing
ALES-GESLA-ZOI and ALES-GESLA-250km) stems from the reduction of the power law and white noise amplitudes in the
residuals. This is in turn caused by improvements of the comparability of TG and altimetry measurements at high-frequency
(i.e. days). We argue that extending the maximal radius selection from 250 km to 300 km to construct the ZOI (as done
for ALES-GESLA-ZOI) increases the low-frequency noise (indicated by $\kappa$). However, with this selection we capture more
altimetry tracks with similar highly correlated high-frequency signals (see Figure 1), which again contribute to sampling
density and reduced white noise. This further substantiates our choice to select SLA within a larger 300 km radius, which is
also supported by observed larger-scale coherency of coastal sea level trends (see section 3.2).

$RMS_{\Delta VLM}$ and trend uncertainty level off at very high thresholds and ultimately increase when only 5% of the data is used (Figure 5a and 5c). We argue that this is mainly related to a decrease in sampling-density of the time series included in the selection: At the 95th percentile, the median sample size (i.e. number of monthly averages in a time series) is 20% smaller that the sample size at the 80th percentile. Robust trend estimates require a minimum of samples, hence, using a reduced number of along-track data time series, even when they show a maximum degree of comparability, yields on a global average decreased trend accuracy ($RMS_{\Delta VLM}$). We thus argue that the optimum threshold identified at about the 80th percentile (of the data sorted by RMS) represents a compromise between data-comparability, as well as sampling-density of altimetry data. We emphasize that there are numerous factors, other than the time period covered, which may contribute to a lack of comparability of SAT-TG and GNSS trends. We further elaborate those in the subsequent discussion section 5.

When setting this optimal threshold to 20%, the ALES-GESLA-ZOI set-up outperforms the other investigated configurations. Figure 4 compares the scatter of estimated $VLM_{SAT-TG}$ against GNSS trends of all datasets. For ALES-GESLA-ZOI, we find lower $VLM_{SAT-TG}$ trend uncertainties and reduced spread of the estimates with respect to the 1:1 line (Figure 4). These results underpin that a refined selection procedure (ZOI) represents the dominant advancement, as this approach outstrips the improvements (in terms of trend accuracy and uncertainty) which are obtained from using different altimeter or TG data combinations.

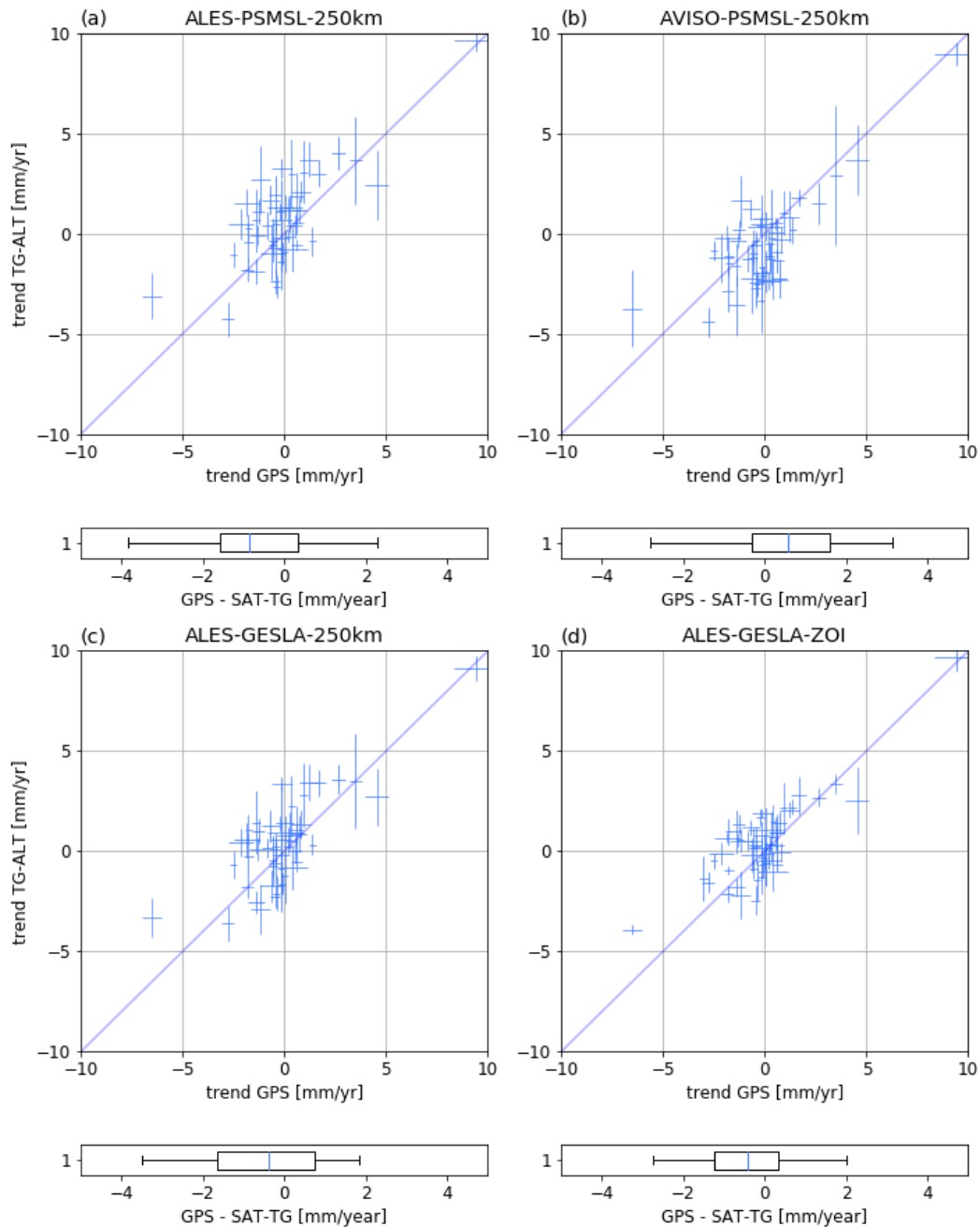

**Figure 4.** Scatter and boxplots compare estimated SAT-TG trends and GNSS trends, as in WM16 Figure 14. a) ALES-PSMSL-250km b) AVISO-PSMSL-250km c) ALES-GESLA-250km d) ALES-GESLA-ZOI (at 20% threshold based on RMS-criterion). Error bars denote the 1 sigma trend uncertainties of the individual estimates.

Figures 5a and 5c illustrate the influence of applying different criteria on the performance of estimated trends. Generally, increasing relative $RMS_{SAT-TG}$ or correlation thresholds yields similar optimal ranges ($\sim 20\%$) for both $RMS_{\Delta VLM}$ or uncertainty of $VLM_{SAT-TG}$ trends and can thus be interchangeably used. At lower relative threshold levels (20-60%), however, application of the RMS-criterion yields slightly reduced $RMS_{\Delta VLM}$ values compared to correlations. Hence, for this set of TGs a SLA-selection based on the minimum $RMS_{SAT-TG}$ generally provides more accurate trend estimates (in terms of $RMS_{\Delta VLM}$). The residual annual cycle criterion only weakly reproduces the improvements provided by the other criteria and is less suited to confine the ZOI. This observation emphasizes the need of matching the data according to the high-frequency comparability (RMS, correlation), because selecting the data based on the residual annual cycle (i.e. low frequency comparability), limits the performance of the estimates (Figure 5c). Considering improvements in the bias of trend differences, we find no significant differences in using different thresholds. In contrast to the improvements in accuracy (as shown in Figure 5), the median $\Delta VLM$ does not converge to a global optimum. Therefore, we discuss the contribution of other factors affecting the comparability of SAT and GNSS in section 5.2.

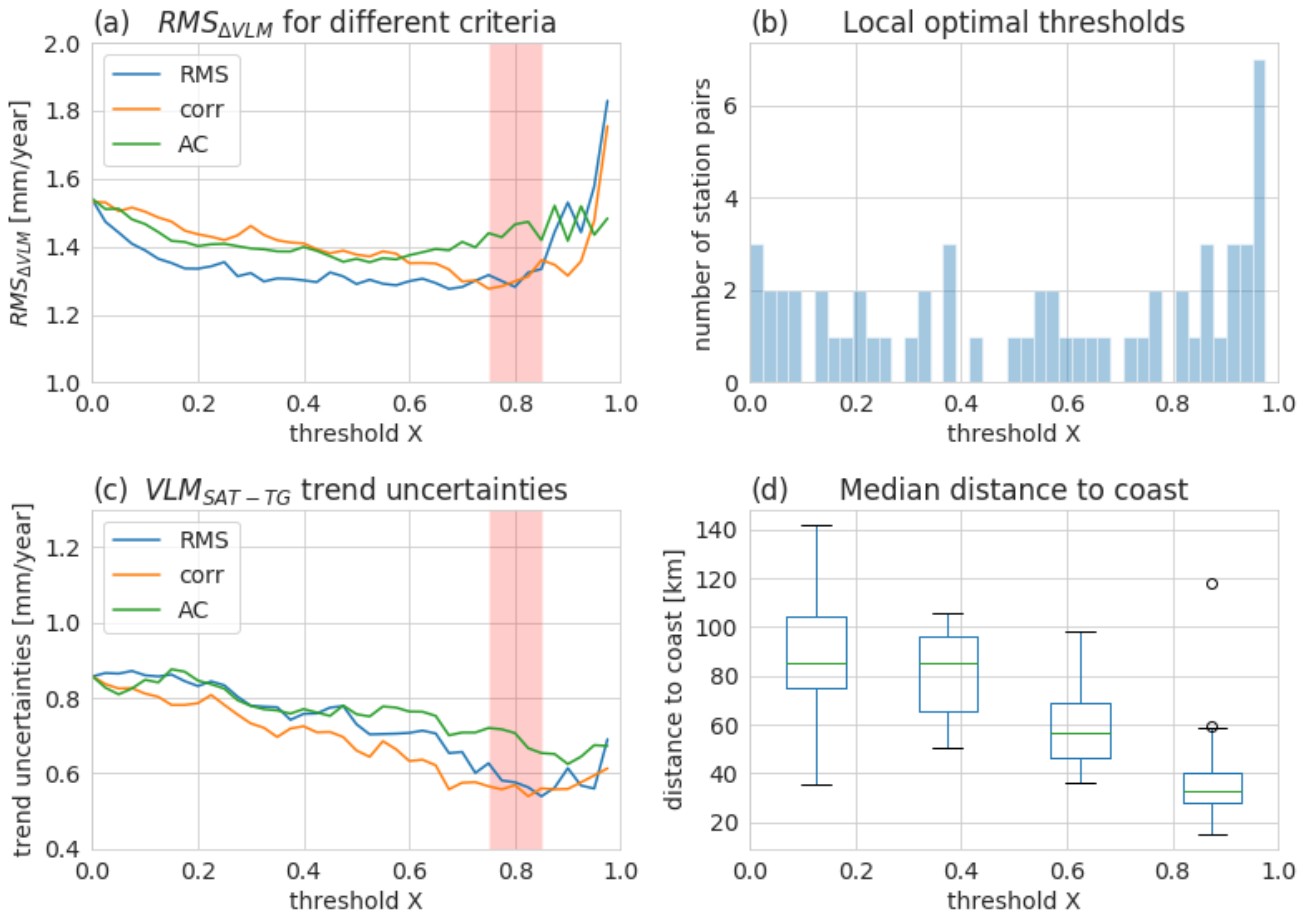

**Figure 5.** Performance of VLM$_{SAT-TG}$ trend estimates for ALES-GESLA-ZOI. a) RMS$_{\Delta VLM}$ for different relative thresholds (step size 2.5%) and different selection criteria: RMS$_{SAT-TG}$ (blue), correlation (red) and residual annual cycle (AC, green); c) same as (a) but for median uncertainty. b) Distribution of best performing relative thresholds for individual stations. The local optimal threshold is defined at the minimum of the absolute difference of VLM$_{SAT-TG}$ and GNSS trends. d) Boxplot shows the distribution of the mean distances to coast for the individual optimum ZOI's as denoted in b). The distances refer to the distributions within the 0-25%, 25-50%, etc. levels, respectively.

## 5  Discussion

The integration of the ZOI primarily reduces the uncertainty of VLM$_{\text{SAT-TG}}$ trend estimates. Over a considerable range of thresholds (80 - 20% of best performing data) trend accuracies do not improve as strongly as the uncertainties decrease. This is in line with Kleinherenbrink et al. (2018), who showed that for a highly correlated sub-set of TGs, increasing absolute correlation thresholds would not significantly reduce the RMS$_{\triangle\text{VLM}}$. We thus strive to understand better why trend estimates do not always improve when selecting highly comparable (w.r.t. TG) or closely located absolute SLA measurements. This question ultimately leads to the discussion of the importance of identifying the small-scale dynamical components of local sea-level variability, given that long-term absolute sea level trends are large-scale signals.

### 5.1  Space and time dependencies of coastal sea level trends

The results presented in Figure 5 and Table 2 denote metrics and performances derived from the global TG-GNSS dataset for ALES-GESLA-ZOI and support an optimal threshold at 20%. It remains to be investigated, whether the described optimum global threshold also reflects the best choice at every coastal site considered. Therefore, we investigate at which relative levels individual VLM$_{\text{SAT-TG}}$ and VLM$_{\text{GNSS}}$ trends estimates yield the smallest absolute deviations. Postulating that the actual VLM at the TG location is linear and perfectly detected by the GNSS station, these thresholds denote the 'local' optimal levels. With this analysis, we aim to better understand the spread of individual optimal ZOIs and what would be the best theoretically achievable RMS$_{\triangle\text{VLM}}$. This analysis also provides a basis to motivate future investigations, in particular to identify systematic factors, which may lead to locally different extents of the ZOI and to improve the accuracy of trend estimates.

Figure 5b displays the distribution of local optimal thresholds for TG-GNSS stations for the ALES-GESLA-ZOI dataset. Note that these estimates are not independent as they are based on prior knowledge of the ground truth VLM from GNSS. Overall, the optimal levels $X$ are broadly distributed from 0 to 0.975. We find highest concentrations between 0.8-0.975, which slightly exceeds the range of the global optimum. At the global optimum itself (0.8, based on correlations), the median distance to coast (of all SLA measurements in a ZOI) is 39.4 km. 25 % of the altimeter observations are within a range of 20 km to the coast, i.e. the region with the most pronounced coastal advancements of the along-track dataset (Passaro et al., 2015).

In contrast to these examples, we find very low local optima for some stations (Figure 5b). Here, local VLM$_{\text{SAT-TG}}$ and GNSS trend differences do not converge to a minimum when increasing the comparability of SAT and TG observations. Accordingly, in these cases, vertical land motion estimates do not necessarily benefit from high coastal resolution of the data, because a low relative threshold is simultaneously linked to a larger-scale selection of SLAs (Figure 5d). At the lower level ranges, for instance at 0-0.2, SLAs have an average distance of 95 km to the coast. Supposing that the sources of these larger scales of coherency of coastal SL trends would be known, a more advanced adaption to these additional factors would further increase accuracy of VLM estimates. An associated ideal selection of trends, based on optimal individual levels shown in Figure 5d would largely reduce to RMS$_{\triangle\text{VLM}}$ to 0.89 mm/year. We emphasize that this constitutes the best RMS, which could theoretically be achieved with our dataset combination, if all of the local optimal levels could be systematically explained. This demonstrates that, albeit there might be room for minor improvements, there is still a strong limitation remaining in bringing the RMS below 1 mm/year.

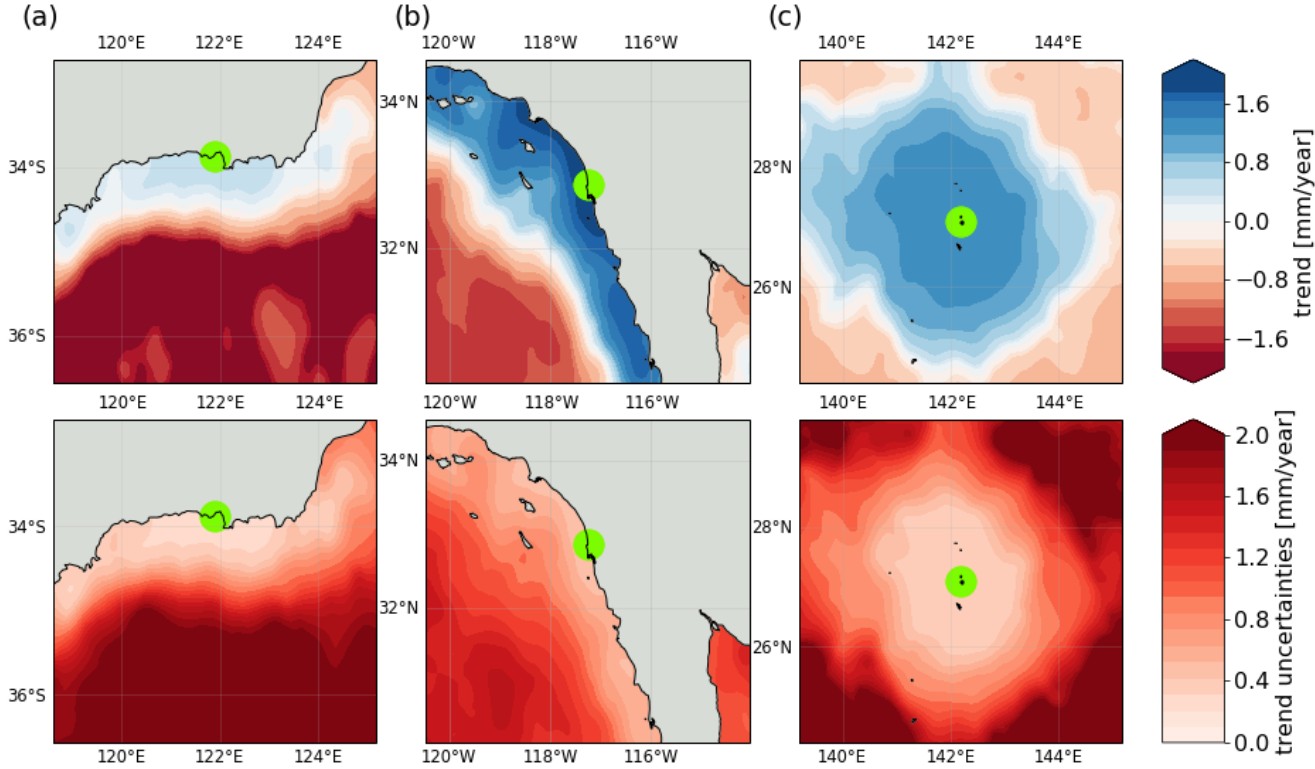

**Figure 6.** VLM$_{\text{SAT-TG}}$ trends (first row) and uncertainties (second row) mapped onto relative correlation levels. The mapping and interpolation method is further elucidated in the Appendix A. We show the same stations (a,b,c) as in Figure 1.

To further shed light on the relationships between dynamical-sea-level-based SLA selection and spacial coherence of trends and uncertainties, we show trend and uncertainty maps (Figure 6) in accordance with those in Figure 1 (displaying maps

of statistical criteria). Here, we map linear trends and uncertainties onto observed levels of comparability defined by the correlation-criterion. We thus compute these VLM$_{\text{SAT-TG}}$ trends over different coastal regions (see further details in the Appendix A). As a result, we observe sharp trend gradients consistent with the degree of comparability: Figure 6a for instance, shows high small-scale variability of trends, because trends off the slope-current-region are detached from the trends in the along-shore continental shelf region. Trends in Figure 6a and 6b project onto the far-reaching along-shore correlations as ob-

served in Figure 1 showing consistent signals over several hundreds of kilometers along the coast. Uncertainty maps further pronounce the importance of the application of highly resolved coastal altimetry data (Figure 6 lower row). These examples show, that at individual locations the use of less comparable SLAs can increase the uncertainty by a factor of three to four. Therefore, for the majority of cases, these results promote using high relative levels of comparability to define the ZOI for trend estimation. However, we also observe that the coherency of trends (highlighted by the strength of absolute trend gradients) can

be differently expressed at different coastal regions.

Bathymetric and coastal properties can cause large discrepancies in responses of coastal sea level variability as they modify the character of the impact of large-scale atmospheric forcing or remote variability from the deeper ocean (Woodworth et al., 2019). Hence, an advanced analysis of SL coherency and the role of bathymetry might facilitate further enhancements of trend accuracy based on SAT and TG. We note that physical origins might, however, not necessarily cause the spread of individual optimal thresholds (Figure 5b). If our assumption, that GNSS-trend estimates perfectly represent the linear trend over the time span of the altimetry/TG records was not met, the shown individual thresholds would erroneously reflect local optima. Ruling out these sources of error is thus a prerequisite to further study physical explanations for different extents of the ZOI.

Next to site-dependent physical factors, the spatial-scales of trend coherency might also depend on the time span of the observations themselves. Global maps of sea level trends, for example, even when derived from two decades of observations, still show distinct pattern of natural/forced variability and thus shade signals of ocean mass or steric contributions (e.g. Stammer et al., 2013). Similarly, coastal sea level trends that are computed in the ZOI are affected by local interannual sea level variability on top of the secular trend. Therefore, the importance to adopt the concept of the ZOI for improving trend accuracy might also be influenced by the actual time span covered by the record.

To investigate this time-scale-dependency, we truncate the $\text{VLM}_{\text{SAT-TG}}$ time series such that we obtain different experimental ALES-GESLA-ZOI sets with maximum record lengths from 10 to 18 years. We repeat the same validation analysis against GNSS trends as in section 4. Figure 7a) encompasses anomalies of the $\text{RMS}_{\Delta\text{VLM}}$ with respect to the mean $\text{RMS}_{\Delta\text{VLM}}$ for a dataset of a specific time scale which is given in Figure 7b (red). The same evolution is shown for trend uncertainties in Figure 7c).

Mean $\text{RMS}_{\Delta\text{VLM}}$ as well as mean uncertainties (which are averaged over all relative thresholds for a specific maximum record length) substantially decrease with increasing record length (Figures 7a and 7c). Both statistics approximately follow the theoretical proportionality of uncertainty and sample size $n$ of $1/\sqrt{n}$ (assuming no serial correlation). The evolution of the $\text{RMS}_{\Delta\text{VLM}}$ anomaly shows that selecting SLAs in a ZOI at high relative thresholds more substantially reduces the $\text{RMS}_{\Delta\text{VLM}}$ on shorter time scales (e.g. 10 years) than on longer time scales (Figure 7a, e.g. at 18 years). At long time scales, the $\text{RMS}_{\Delta\text{VLM}}$ anomalies do not significantly improve between the 80% and 20% thresholds, which we also observe in the previous analysis in Figure 5a. We argue that the transition time scale where the improvements of $\text{RMS}_{\Delta\text{VLM}}$ flatten (14-16 years), marks when the high frequency coastal sea level dynamical variability is superseded by dynamics producing large-scale sea level trends. In other words, this is the time scale in which coastal sea level trends start to merge with the offshore trends. The tendency of increasing spatial scales with time is also reflected by the increasing distances to coast of the measurements for an optimal ZOI at a specific time scale (Figure 7b). The time-scale-dependency could explain the mismatch of trend accuracy and uncertainty improvements when using higher levels of comparability. This is also supported by Kleinherenbrink et al. (2018), who showed little sensitivity for SAT-TG combinations which had minimum lengths of 15 years.

The same evaluation for the dependency of uncertainty on time and level of comparability $X$ demonstrates that using the ZOI nearly constantly improves trend uncertainties at any time scale. Hence, even though spatial scales of trend coherency might increase with time, an ideal match of altimetry and TGs should be based on a ZOI.

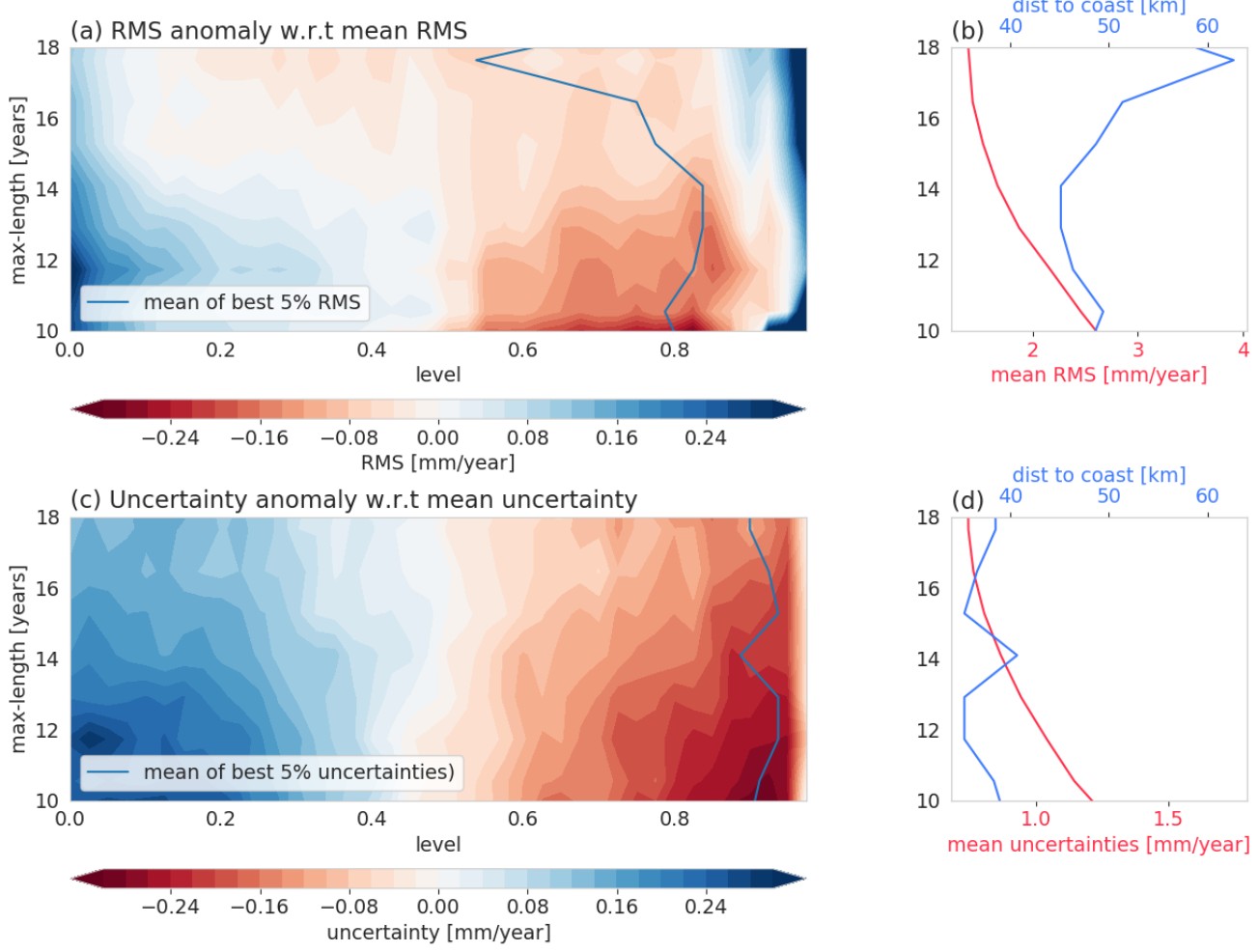

**Figure 7.** Time and space dependencies of trend uncertainty and accuracys: a) Evolution of the RMS$_{\Delta\text{VLM}}$ anomaly (SAT-TG vs. GNSS trend) for subsets of ALES-GESLA-ZOI, depending on a relative threshold $X$ (x-axis) and a maximum record length (y-axis). RMS$_{\Delta\text{VLM}}$ anomaly is defined as the departure from the mean RMS$_{\Delta\text{VLM}}$ (shown in b) averaged over all thresholds $X$ for a specific maximum record length. In b) we also show in blue the mean distance to coast of the measurements, associated with the average of the best 5% ZOI-levels per time scale, shown in a). c,d) Same as a,b) but for uncertainties.

## 5.2 Systematic errors

VLM estimates from different datasets (e.g. AVISO-PSMSL-250km and ALES-GESLA-ZOI) are biased compared to trends inferred from GNSS observations. Based on Monte-Carlo simulations (see appendix Figure B1) we argue that these biases are significant for most of the dataset combinations (ALES-PSMSL-250km and AVISO-PSMSL-250km, ALES-GESLA-ZOI). In the following, potential sources for these biases will be discussed.

Next to the record-length (see section 5.1), systematic errors critically affect the accuracy of the SAT-TG technique and can have strong systematic effects on the trend differences. Limiting factors for VLM determination from both SAT-TG and GNSS observations are the accuracy and uncertainty of origin and scale of the reference frame (see WM16, Collilieux and Woppelmann (2009); Santamaría-Gómez et al. (2012)), which cannot be realized yet at the required accuracy level (Bloßfeld et al., 2019; Seitz et al., submitted).

Moreover, as mentioned before, the multi-mission calibration applied (MMXO) reduces intermission biases as well as regionally coherent systematic errors, but does not feature a calibration against TG. The median bias identified for ALES-GESLA-ZOI could be affected by a drift of the mission used as reference. In contrast, the AVISO dataset does not include time-dependent intermission biases and might therefore be additionally influenced by systematic effects of e.g. Envisat or Sentinel-3a (Dettmering and Schwatke, 2019).

Next to to altimeter bias drift, nonlinear VLM from contemporary mass redistribution (CMR) changes were shown to cause differences between $VLM_{SAT-TG}$ and $VLM_{GPS}$, due to the different time periods covered (e.g. Kleinherenbrink et al., 2018). Using GRACE (Gravity Recovery and Climate Experiment) observations Frederikse et al. (2019) demonstrated that associated deformations can cause VLM trend on the order of 1 mm/year. Therefore, they introduced a new method to reduce $VLM_{GPS}$ by GIA and CMR signals to minimize their associated induced extrapolation biases. Kleinherenbrink et al. (2018) incorporated

nonlinear VLM from CMR to assess the corresponding trend differences between $VLM_{SAT-TG}$ and $VLM_{GPS}$. They expose that $VLM_{SAT-TG}$ estimates are lower than $VLM_{GPS}$ in many parts of North America and Europe and higher in subtropical/tropical regions as well as Australia and New Zealand (refer to Figure 9 in Kleinherenbrink et al. (2018)). Because northerly regions like North America are affected by dynamic changes in CMR, GNSS observations which cover shorter and more recent time spans than satellite altimetry detect stronger uplift signals. For a set of 155 TG-GNSS pairs, integration of these signals

slightly reduced the median bias from -0.14 mm/year to -0.07 mm/year, but had no significant effect on RMS. Given that most of the TG-GNSS stations used in this study are located in Europe, North America and Australia, CMR might as well alleviate the negative trend bias of ALES-GESLA-ZOI. Therefore, extending the validation platform, not only by using other homogeneous GNSS observations, but also GRACE and GIA estimates would strongly support identification and mitigation of such systematic errors.

## 5.3 Comparison with previous results

Based on optimal relative thresholds, we estimated an $RMS_{\Delta VLM}$ between SAT-TG and GNSS trends of 1.28 mm/year and a median uncertainty of 0.58 mm/year at 58 sites. Our approach of combining along-track altimetry, high frequency TG data and a refined SLA selection-scheme improves the performance of VLM estimation compared to using gridded altimetry products and constant spatial SLA averages (WM16: $RMS_{\Delta VLM}$: 1.47 mm/year; uncertainty 0.80 mm/year). Other studies further

emphasized the importance of spatial resolution in coastal zones, considering the decreasing temporal and spatial scales of sea level variability in such areas. With the focus on coastal sea level trends, Cipollini et al. (2017) demonstrated that the along-track X-TRACK product contained much more valid data close to the coast than AVISO, not only due to the spatial down-sampling in AVISO, but also to less adapted coastal processing. Here, we tackle both issues by implementing an ad-

vanced coastal along-track altimetry product. Because we find that much of the observed high-performing altimetry data has
a close vicinity to the coast, our results underpin that along-track data is the best choice for coastal sea level trend estimation,
and substantiate the results of Kleinherenbrink et al. (2018).

Accuracies of estimated $VLM_{SAT-TG}$ expressed by $RMS_{\Delta VLM}$ are in the order of Kleinherenbrink et al. (2018)'s result of
1.20 mm/year. These results can, however, not unequivocally compared due to different validation settings. We extend their
analysis by investigating a variety of other criteria of comparability and find that the $RMS_{SAT-TG}$ of the differenced $VLM_{SAT-TG}$
time series provides the most robust estimates compared to correlations or residual annual cycle. Our results also propose that
increasing the radius of selection denotes another improvement for VLM estimates. Practically, the approach of using absolute
thresholds, which was put forward by Kleinherenbrink et al. (2018) almost halved the number of considered stations from 294
to 155, when setting an absolute correlation threshold to 0.7. Applying relative thresholds, facilitates the estimation of trends at
lower correlated stations, which would be rejected otherwise. This is crucial, because it was frequently shown, that correlations
between altimetry and TGs are highly variable across the globe (WM16). Hence, we maintain the main advantage of using
TGs for VLM estimation: The large global distribution compared to continuous GNSS-measurements.

We highlight that the SAT-TG estimates are not only limited by the broad spectrum of error sources, ranging from systematic
to correction errors, such as the residual long period tides remaining in the TG time series, which all contribute to the error
budget of the estimates. Another factor is the possible nonlinearity of the VLM itself, which strongly hampers the comparability
with measurements from other geodetic techniques, when sampled over different time spans. Thus, addressing this issue in
SAT-TG time series could represent a further crucial improvement of the application.

## 6 Conclusions

We investigate potential improvements of combining altimetry and TGs for coastal vertical land motion estimation. The inno-
vations of our approach are twofold: (1) For the first time, we exploit a global network of high frequency TG data (GESLA)
and dedicated coastal altimetry (ALES) to determine VLM at a variety of co-located GNSS stations. Secondly (2) we define a
Zone of influence, to identify coherent zones of coastal SL variability which optimizes the combination of altimetry and TGs.
We rate improvements of both innovations against various SAT-TG datasets, which are comprised of along-track and gridded
altimetry, as well as high (daily) and low-frequency (monthly) TG combinations.

Combining high frequency TG with coastal altimetry data (ALES-GESLA-250km) yields modest improvements of trend
accuracies, compared to a monthly gridded or monthly along-track combination, when averaging SLAs in a radius of 250
km. The high spatio-temporal resolution of the data, however, provides the foundation to identify coherent zones of sea level
variability. We confine a Zone of Influence by using relative thresholds of comparability based on $RMS_{SAT-TG}$, correlation
and residual annual cycle of the altimetry and TG timeseries. We identify a global optimal threshold, when selecting 20% of
the data with the lowest $RMS_{SAT-TG}$. At this threshold, validation against GNSS velocity estimates (at 58 stations) yields a
$RMS_{\Delta VLM}$ of $VLM_{SAT-TG}$ and $VLM_{GNSS}$ differences of 1.28 mm/year with a median formal uncertainty of $VLM_{SAT-TG}$ trends
of 0.58 mm/year. This refined selection method improves trend accuracy by 15% and uncertainty by 35% compared to the

250 km-average selection. The smaller degree of improvements of trend accuracy compared to uncertainty is explained by the increasing space-scales of sea-level trend components with progressing time scales. We show that in many cases, capturing small scale features of coastal sea level variability within few tens of kilometers from the coast is vital for $VLM_{SAT-TG}$ estimation and constantly reduces trend uncertainty of the estimates. We thus promote using relative levels of comparability and dedicated coastal altimetry matched with high-frequency TGs to confine ZOIs and increase the number of VLM estimations along the global coastline with improved uncertainty.

*Data availability.* ULR6a GNSS trend estimates are obtained from the data assembly centre SONEL (Systeme d'Observation du Niveau des Eaux Littorales, https://www.sonel.org/-Vertical-land-movement-estimate-.html?lang=en, Santamaría-Gómez et al. (2016)). GESLA tide gauge data are available at http://www.gesla.org (Woodworth et al., 2016) and PSMSL data at https://www.psmsl.org/data/obtaining/ (Holgate et al., 2013). ALES along-track data are processed at DGFI-TUM (https://www.dgfi.tum.de/en/) with OpenADB (https://openadb.dgfi.tum.de). Averaged DT-MSLA AVISO gridded altimetry data are obtained from https://www.aviso.altimetry.fr.

## Appendix A: Methods

In Figure 6 we map linear $VLM_{SAT-TG}$ trends and uncertainties onto observed levels of comparability set by the correlation-criterion. First, we group observed SLA time series in 0-20th, 20-40th, 40-60th etc. percentile-ranges, sorted by their correlations with the TG time series. Then, we merge the altimetry time series for each group and calculate their associated $VLM_{SAT-TG}$ trends. The resulting $VLM_{SAT-TG}$ trends are hereinafter defined on the altimetry tracks, categorized by the aforementioned groups of comparability. To better illustrate the different zones of coherency the trends are interpolated onto a regular grid (100x100 nodes i.e. 6 km resolution) and thus smoothed as seen in Figure 6. We use linear radial basis functions to interpolate the data.

## Appendix B: Significance of median biases

To gain a better understanding of when the $VLM_{SAT-TG}$ and $VLM_{GNSS}$ difference distributions are significantly biased we create a Monte-Carlo experiment to check the $H_0$ hypothesis: 'the median of the distribution is not significantly different from zero' (with alpha=0.025). Therefore, we generated a bootstrapped distribution of random medians, which are derived from 20000 individual sub-sets of size 52 (the number of TG of our dataset), which are randomly drawn from normally distributed values with a standard deviation of 1.5 mm/year (according to the RMS of AVISO-PSMSL-250km) and zero mean.

Figure B1 shows that the biases of the datasets ALES-PSMSL-250km and AVISO-PSMSL-250km exceed the 2.5 and 97.5 percentiles of the sampled distribution (average of absolute bounds: 0.512 mm/year). This means that in less than 5 out of 100 cases, we would obtain such biases by chance, which supports the significance of these biases. We highlight that this is a purely statistical analysis, which cannot account for any of the errors from corrections, adjustments, drifts etc. introduced in the altimeter and GNSS.

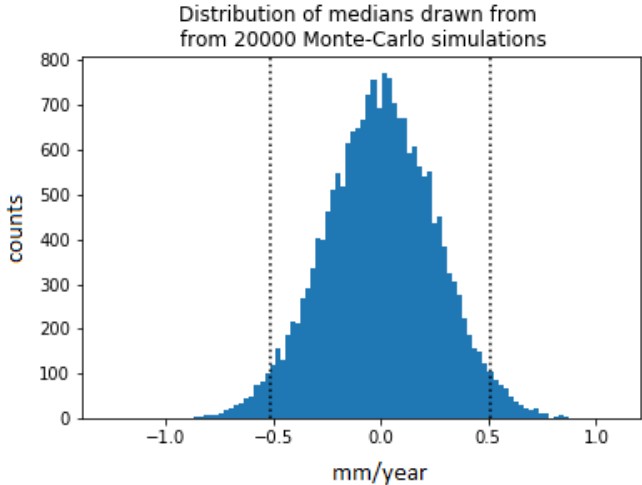

**Figure B1.** Histogram of median values of randomly sampled sub-sets. The sub-sets consists of 52 samples (according to the number of TGs in AVISO-PSMSL-250km) and are randomly drawn from normally distributed values with zero mean and a standard-deviation of 1.5 mm/year (according to the RMS of AVISO-PSMSL-250km). Dashed lines mark the 2.5 and 97.5 percentiles of the distribution.

*Author contributions.* J.O. and M.P. conceptualized and designed the study. J.O. wrote the manuscript and is the author of the full software code used in this study. M.P. is the author of the ALES retracking algorithm and mentored the work of J.O.; C.S. and D.D. are responsible for the altimetry database organisation and the data structure. L.S. provided assistance in the use of GNSS data. F.S. provided the basic resources
making the study possible and coordinates the activities of the institute. All authors read and commented on the final manuscript.

*Competing interests.* The authors declare that they have no conflict of interest.

*Acknowledgements.* This work was funded by the Deutsche Forschungsgemeinschaft (DFG) (grand agreement 411072120) and the Technical University of Munich (TUM) in the framework of the Open Access Publishing Program. We thank the data-providers GESLA, PSMSL, SONEL and AVISO for the opportunity to use their products. We thank Sergiy Rudenko, Ashwita Chouksey and Michael Hart-Davis for
their help and comments. We are very grateful for the comments of the reviewers Alvaro Santamaría Gómez and Christopher Watson, which strongly improved the manuscript.

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
