# Peer review of "The Zone of Influence: Matching sea level variability from coastal altimetry and tide gauges for vertical land motion estimation"

_Ocean Science, 2020_

## Referee Comment (RC1) · Alvaro Santamaría-Gómez (Referee) · 18 Jun 2020

This paper addresses the methodology of estimating vertical land motion (VLM) from the combination of satellite altimetry (SAT) and tide gauge (TG) observations. The work by J. Oelsmann, M. Passaro et al. builds upon earlier studies concerning the selection of the most suitable SAT observations that show high temporal correlation with high-frequency TG observations. In addition, and contrary to past studies, they use dedicated coastal "retracked" along-track observations to reduce the VLM differences with respect to co-located GNSS VLM estimates, which are taken as ground truth.

My feeling is that this paper is a significant technical contribution to the estimation of

coastal VLM from altimeters and tide gauges, which is a relevant topic for the journal. I generally agree with the authors that advances in this field call for better consistency of the sea-level observations from tide gauges and altimeters. To reach this goal the authors focus on using altimeter observations closer to the coast and high-frequency tide gauge data. The authors show that areas of high consistency between both datasets, which they call "zone of influence" or ZOI, can be defined based on different statistical criteria. The comparison of SAT-TG VLM estimates against GNSS VLM estimates is improved, but the typical differences between both VLM estimates are still much larger than their respective formal errors. This indicates that there are still missing pieces to be accounted for in the VLM estimates from SAT-TG or GNSS or, likely, in both.

Below a few minor comments that hopefully will improve the quality of the paper:

Abstract: ZOI should be defined in the abstract, if space allows it.

L24&59: Many thanks for citing my 2017 paper, but there is no need to add it twice to the reference list. 2017a/b should be 2017.

L62: change accuracy by precision

L271-273: "To confine the ZOI, we select subsets of the data containing the best-performing statistics (i.e. highest correlation, lowest RMS SAT-TG or residual annual cycle) above the Xth-percentile according to the distribution of the statistic S in a 300 km radius around the tide gauges." » It is not clear whether the TG and SAT series were detrended and deseasoned before comparing them. If the TG and SAT series were not deseasoned and the seasonal variation is prominent in the series, then there is the risk that the three metrics are telling us almost the same thing, that is, the impact of the amplitude and phase differences between the seasonal signals in both series. The correlation and the RMS of the differences may be more representative from deseasoned series, as done in past studies. In addition the authors fit the seasonal variation together with the linear trend in the SAT-TG series, i.e., the residual seasonal variation may play a minor role in the estimated VLM and its uncertainty. I'm not sure if this is

what the authors intended. It may not have a significant impact on the selected ZOI areas, but the authors would be at least using more independent criteria.

L367: The authors take the GNSS VLM as ground truth, and that is fine, but are the formal VLM errors similar among the TG and GNSS stations, respectively? Formal errors can provide valuable information for the VLM validation and this should be accounted for when assessing the VLM differences (WRMS instead of RMS for instance).

L377-380: the differences between the SLA trends in ALES and AVISO are quite significant. From Table 2 it appears that the median SLA value from AVISO is 1 mm/yr higher than that from ALOS (also seen in Fig. 4). Is this correct? If so, how would you explain this difference?

L386: "absolute median bias of trend differences" can be confused with the median of absolute VLM differences. I suggest changing this by "median of the VLM differences" or similar here and elsewhere.

L400-401: Some comments on results shown in Table 2: the unweighted RMS from GESLA is smaller than that from PSMSL, but this is probably because the median value is closer to zero. The weighted standard deviation or any other measure of dispersion (interquartile range) that does not include the mean/median value would be more appropriate here. Also the formal VLM rate uncertainties are higher with GESLA than with PSMSL, even with a spectral index slightly closer to zero. This means the noise (especially the power-law variance) of the residual series (trend and seasonal variations removed) in the GESLA VLM series is larger than that from PSMSL or that the GESLA series are significantly shorter or less complete. In that case, the choice of using GESLA instead of PSMSL would need better argumentation. There may be also a TG trend bias between GESLA and PSMSL of around 0.3 mm/yr. This is probably not significant, but it may be worth discussing.

L421-422: the power-law variance may have also changed, maybe producing a significant improvement of the VLM formal errors.

L429-436: The discussion in this paragraph is not very clear and would require improvements. We only need a single SLA series to estimate VLM from SAT-TG. This SLA series can be obtained using different strategies as the authors have discussed: spatial averaging/filtering of SLA data, the single most correlated SLA series, the single closest SLA series, etc. The more similar the selected SLA series is to the TG series, i.e., the smaller the SAT-TG differences (again excluding the seasonal variations that are captured by the model fitted to the SAT-TG series), the more precise VLM will be obtained in terms of formal error. This is a metric very easy to interpret. Note that interpreting how the SAT-TG VLM values compare to the GNSS VLM is much more complex and is, in general, not a strong criterion given the large differences between both. The smaller quantity of averaged SLA should not be blamed if they represent increasingly consistent SLA series with respect to the TG series. A different explanation for the bad results with >80% thresholds could be that the RMS metric is not telling us whether the SAT and TG series are more similar, especially if the seasonal signals were included in the RMS as per my comments above. In addition, the RMS alone is not directly tied to the autocorrelation of the SAT-TG series (i.e., the spectral index), which is another important metric to assess the consistency of the SLA and TG series.

L466-478: the discussion here is interesting, but an important point should be stated more clearly. The optimal ZOI in Fig. 5 a&c was retained by assessing the consistency between SAT and TG series. On the other hand, the local optimal ZOIs in Fig. 5b are defined by comparing SAT-TG and GNSS VLM estimates. It is therefore not surprising that different optimal local ZOIs are obtained and that there is not an optimal threshold that fits well all sites. In addition, imposing the GNSS VLM as a criterion to the computation of the SAT-TG VLM would remove its independent nature.

L487-488: I guess you derived the 3 to 4 inflation factor from the color range in Fig. 6, but Table 2 actually shows that using different SLA data does increase VLM uncertainties by less than 10% (comparing ALES-PSMSL to AVISO-PSMSL).

Alvaro Santamaría, June 2020

**[OSD](https://doi.org/10.5194/os-2020-29)**

Interactive
comment

---

## Referee Comment (RC2) · Christopher Watson (Referee) · 21 Jul 2020

This manuscript by Oelsmann et al provides a contribution to the improved determination of vertical land motion (VLM) at tide gauge locations using satellite altimetry. The work advocates for a refined approach to selecting valid altimeter data in a variable 'zone of influence' around a specific tide gauge to improve the accuracy and precision of VLM estimates. The team make incremental advances using a coastally retracked altimeter dataset and tide gauge data with various temporal resolutions. The focus of the paper has a rich history in the literature and this technical contribution provides a worthy contribution to progressing the method and ultimately achieving improved VLM

estimates around the coast. The demand for improved VLM is clear – VLM is a vexing issue with manifold impacts across the scientific and the broader community.

The manuscript is well presented, well crafted and generally presents a compelling and comprehensive case worthy of publication in this outlet. Noting my support for this manuscript, I have a few substantive issues that I feel would benefit from consideration and discussion/revision in the paper. I present these issues below, followed by a list of technical clarifications and minor suggestions/corrections that may benefit the manuscript.

Main Issues:

Central to the manuscript is the comparison of various altimeter minus tide gauge differences against a single 'ground truth' GPS solution. Significant differences in the solutions derived from different variants (e.g. Fig 4b) are evident, highlighting the sensitivity of the problem. This suggests that fundamental issues remain to be understood in one or more of the constituent components used (i.e. any one or all of ALT/TG/GPS). Relating to this point and notwithstanding my remarks above regarding the relevance of this work, my sense from reading the manuscript is that the authors pay too little attention to discussing a few key issues that I argue should otherwise appear in a paper such as this:

1) Despite the technique itself originally emerging at the time investigators were using tide gauges (and thus VLM) to validate the stability of the altimeter measurement system, the authors do not seem to acknowledge the potential for regionally correlated error within the altimeter – brief mention of the importance of this point in the heritage (and current use) of the technique is also surprisingly lacking. Rather, the authors make the bold assumption of 'no instrumental drifts' (pg 3, ln 75) as the sole mention of possible systematic error in the altimeter record. I note the authors exclude TOPEX/Poseidon from the coastally retracked dataset given waveforms are presently unavailable, however, it is included in the AVISO product. The well-known issues associated with TOPEX (brought to light using this technique and later confirmed) put the issue into focus. Further, inspection of solely orbit differences in a regional context places even greater emphasis on the issue (e.g. Couhert et al, ASR (2015) who report on the significant challenge of achieving 1 mm/yr orbit stability at regional scales). The use of a consistent orbit across all missions in this current paper (which I assume are the GFZ Ver13 SLCCI orbits?), will assist to mitigate this effect but it is likely that mission-specific, spatially-correlated 'whole-of-system' drift will remain in the residuals and undoubtedly make at least a contribution to the differences observed in the current paper. I contend that 'no instrumental drifts' is somewhat of a misnomer in the context of whole-of-system altimetry at regional scales. Not mentioning these broad issues nor the heritage of the ALT-TG technique in this regard would be disappointing for a paper such as this.

2) A second issue of less significance relates to the treatment of the tides in the high frequency tide gauge data. The authors attenuate the tide with a 40-hour loess filter (pg 8, ln 217). Clarification and further defence of the filter strategy would be beneficial here (e.g. both are likely small, but I wonder what % of *non*-tidal variance is attenuated by the filter, and what % of *tidal* variance at longer periods is retained (but removed from the altimeter with its model-based correction)). The comment regarding the temporal resolution of the DAC used for the high rate TG data felt ambiguous here (ln ∼219), and should be clarified with respect to consistency with the filtering approach. The magnitude of the difference in tides between (coastal) altimetry that uses a global model and at the tide gauge is an important issue that warrants at least a mention in the discussion.

3) Finally, the improvement observed between along-track altimetry v gridded altimetry is presented in the context of 'coastal altimetry' driving the improvement. I'm very pleased to see these results however I feel that conclusion needs some further defence. From my understanding of Fig 5d, the lower quartile of the data commences at just under 30 km from the coast (and increases from there). Discussion of Fig 5b in the text

(pg 22, ln 470) refers to a much smaller subset in which the mean distance was 33 km. Given the benefit of retracking is 'most pronounced in the last 20 km' the authors should further evidence what % of their data is within this threshold and thus further elucidate whether it is simply the influence of along-track data versus its retracked equivalent. To be clear, I am not asking for the former to investigated in the same way – the benefits of the retracking in general terms are clear – I just wish to ensure the conclusions are appropriately elucidated.

Technical Issues:

It is unclear throughout if the RMS statistics computed against the GPS time series take into account the GPS uncertainties? I expected to see WRMS mentioned rather than RMS.

In section 2.1, I assume the GFZ Ver 13 SLCCI orbits are used homogeneously for all missions, but this is only inferred. I suggest adding detail to Table 1 for clarity and citing the relevant Rudenko et al paper.

I was pleased to see the data has been cross calibrated using the MMXO approach (pg 6, ln 165). Some additional detail here would be beneficial: I assume this provides a location-specific and mission-specific radial bias, computed using the *identical* dataset (Table 1). Some further detail on this would be useful here. Further, some comment on the relative trends observed (and applied?) between the Jason-series and ERS/Envisat missions (using the MMXO approach) is warranted here.

Regarding the pole tide in Table 1, could the authors confirm they only apply the solid Earth component of the pole tide (noting the pole tide that is present in the GDR combines the ocean plus solid earth component, hence needs to be multiplied by h2/(1+k2).

Regarding the ocean tide in Table 1, it might be worth adding a paragraph to further discuss the long-period tides that are/are not considered in the altimeter data. The current treatment of the tide in the high frequency dataset at least should retain any

long period constituents – confirmation that is the case with the altimetry would also be beneficial.

Regarding the cubic interpolation to the hourly tide gauge data, I'm assuming you don't use this across outages larger than some threshold?. Worth clarifying.

In Figure 2, this may be more beneficial to show the data in panel 3 over the data in panel 2. Then, in the now vacant third panel, show the same but with the annual and semi-annual terms estimated and removed. Note the caption reports "SAT minus TG" but the y-axes report "TG-ALT".

In Figure 4, I found Fig4b and d difficult to interpret. I feel they need to be introduced earlier and/or more comprehensively explained as I felt it took me a few reads to get the point.

Minor Issues:

The word 'frequent' is used throughout in the context: 'high frequent data'. I suggest this should be 'high frequency'.

The terms accuracies and uncertainties are used throughout when singular is arguably more appropriate. E.g. ln 11 in the abstract.

Suggest a search and replace for units – some (e.g. km) do not have a space before-hand after the quantity.

L15 p1. Perhaps increasing instead of progressing? Perhaps . . .the spatial scales of the coherency in coastal sea level trends increase.

L20, p1. . . .as the sea level rise signal itself. . .

L21, p1. . . .can regionally account for a large fraction of the. . .

L35, p2. I had expected TG instability to at least be mentioned here. I know it comes later, but consider a mention here.

L40, p2. "global" trends range 1 to 3 mm/yr. Do you really mean "global"? If you do then I feel this could be made more specific to a time period of interest as it feels too vague.

L54, pg2. . . .solutions against those using measurements. . .

L59, pg3. "considerable" isn't the right word here, consider change.

P3 – See main issue #1.

L80, p3. "synergistic applications" isn't quite right, consider change.

L87, p3. . . .further develop. . . instead of "carry on" the latest progress? Suggest characterisation and quantification instead of detection?

L96, p4. "were achieved" c.f. what?

L108, p4. suggest: . . .however, they reported insignificant improvements of. . .

L126, p4. no comma needed after processes.

P6 – See main issues 2 and 3.

L190, p7. The PSMSL constitutes. . .

L191, p7. . . .for most sea-level research.

L222, p8. low latitude maybe ambiguous for some readers. Do you mean nearer to the equator or poles? I note you use high-latitude later.

L240, p8. I'm unsure that "matching" is the correct word. "Differences formed using" perhaps?

L252, p9. . . .the capability to compare. . . no comma after altimetry.

L264, p9. duplicate as

L310, p12. performance not performant

L330, p14. "next to SLA correction" could be significantly improved.

L~340, p14. Some comment from the authors regarding whether they consider PL+WN more appropriate than GGM for example would be useful.

L349, p14. needs "ellipsoid" inserted after TOPEX/Poseidon.

Figure 3: The distribution is unavoidably clustered. Does residual VLM show any spatial patterns? L370, p15.is "also" needed?

L399 p16. Suggest improve "which stronger deviate".

L402, p16. This is the opposite sign however. . .

L435 p18. assuming issues with e.g. tide model errors, the method presented will likely have a optimum threshold that still enables sufficient spatial averaging to mitigate that systematic effect as much as possible. See main issue 2.

L439, p18. Is bisector the correct term? 1:1 line?

L441, p18. Unclear what you mean by outstrips improvements induced by. . .. Clarify.

Figure 4 – Comment error bars are 1 sigma?

L447, p20. The residual annual cycle criterion. . .

L450, p20. I tend to agree with the statement, but what about the bias in the accuracy as arguably the low frequency component may lead to larger biases in deltaVLM (as well as scatter). A comment maybe useful here.

Figure 5. In the legend, unclear what AM is. I assume something to do with Annual, but needs clarification in the caption.

L455, p22. estimates do not always

L458, p22. See main issue #1.

L460, p22. Is average the best word here. I understand your point, but statistically

speaking? Perhaps denote metrics of performances derived from the global dataset?

L463, p22. at every coastal site considered.

L469, p22. "large" in inadequately defended. See issue #3.

Section 5.1 was challenging to read, Fig 4b and d were difficult to interpret. I suggest some further elaboration.

L494, p23, no comma needed after we note.

L503, p24. I understand your point here but I feel more emphasis should be placed on the sensitivity of the ALT and TG to the same phenomena. If both were sensing 100% the same signal, the time span is not as critical. Again, I come back to main issue #1 here.

L535, p26. Quantify "much of" and "close vicinity to the coast". See main issue 3.

L538, p26. . . .not be unequivocally

L560, p26. Are these statistics medians?

L565, p26. . . .and dedicated coastal altimetry. . . confine ZOIs and increase. . . . . .the global coastline which improved uncertainty.

---

## Author Comment (AC1) · 27 Aug 2020

We thank Alvaro Santamaría-Gómez for his very constructive and positive comments, which are extremely valuable for improving the manuscript. In response to the comments, we performed additional computations and integrated the associated results. In order to refine some of our messages and to make the manuscript clearer, we reformulated explanations or interpretations of the data at several positions in the text.

On behalf of all authors,

Julius Oelsmann

[Figure]

Please also note the supplement to this comment:
https://os.copernicus.org/preprints/os-2020-29/os-2020-29-AC1-supplement.pdf

**Supplement:**

Review response

To Alvaro Santamaría-Gómez, August, 26 2020

*We thank Alvaro Santamaría-Gómez for his very constructive and positive comments, which are extremely valuable for improving the manuscript. In response to the comments, we performed additional computations and integrated the associated results. In order to refine some of our messages and to make the manuscript clearer, we reformulated explanations or interpretations of the data at several positions in the text.*

*General note: We changed the Envisat mission data version from V2.1 to V3 (for the ALES altimetry data). This influences all associated dataset combination (ALES-PSMSL-250km, ALES-GESLA-250km and ALES-GESLA-ZOI). Because the statistics are not significantly altered, the key messages of the study remain. All statistics and plots are updated accordingly (i.e. Figure 1,2,4,5,6,7 and table 2).*

*We use italic formatting to answer the comments. Existing text is marked in* blue *and changes in the text are highlighted in* red*. All line numbers refer to the originally submitted version.*

**This paper addresses the methodology of estimating vertical land motion (VLM) from the combination of satellite altimetry (SAT) and tide gauge (TG) observations. The work by J. Oelsmann, M. Passaro et al. builds upon earlier studies concerning the selection of the most suitable SAT observations that show high temporal correlation with high-frequency TG observations. In addition, and contrary to past studies, they use dedicated coastal "retracked" along-track observations to reduce the VLM differences with respect to co-located GNSS VLM estimates, which are taken as ground truth. My feeling is that this paper is a significant technical contribution to the estimation of coastal VLM from altimeters and tide gauges, which is a relevant topic for the journal. I generally agree with the authors that advances in this field call for better consistency of the sea-level observations from tide gauges and altimeters. To reach this goal the authors focus on using altimeter observations closer to the coast and high-frequency tide gauge data. The authors show that areas of high consistency between both datasets, which they call "zone of influence" or ZOI, can be defined based on different statistical criteria. The comparison of SAT-TG VLM estimates against GNSS VLM estimates is improved, but the typical differences between both VLM estimates are still much larger than their respective formal errors. This indicates that there are still missing pieces to be accounted for in the VLM estimates from SAT-TG or GNSS or, likely, in both. Below a few minor comments that hopefully will improve the quality of the paper:**

**Abstract: ZOI should be defined in the abstract, if space allows it.**

*We changed L7-8 to:*
*'To improve the coupling-procedure, a so-called 'Zone of Influence'* (ZOI) is defined, *which confines*  *coherent zones of sea level variability on the basis of relative levels of comparability between tide gauge and altimetry observations.'*

**L24&59: Many thanks for citing my 2017 paper, but there is no need to add it twice to the reference list. 2017a/b should be 2017.**

*Fixed!*

**L62: change accuracy by precision**

*Changed!*

**L271-273:** *(comment separated by author) part1:* **"To confine the ZOI, we select subsets of the data containing the best performing statistics (i.e. highest correlation, lowest RMS SAT-TG or residual annual cycle) above the Xth-percentile according to the distribution of the statistic S in a 300 km radius around the tide gauges."** **» It is not clear whether the TG and SAT series were detrended and de-seasoned before comparing them. If the TG and SAT series were not de-seasoned and the seasonal variation is prominent in the series, then there is the risk that the three metrics are telling us almost the same thing, that is, the impact of the amplitude and phase differences between the seasonal signals in both series. The correlation and the RMS of the differences may be more representative from de-seasoned series, as done in past studies.**

*We only detrended the data (and removed an offset) before matching. We agree with the reviewer that without de-seasoning the independency of the metrics is reduced as they are all also influenced by the consistency of annual cycle signal of both time series. We did not de-season the data in the first place, as we use high-frequency data and assume that the spatial coherency (and thus the extent of the ZOI) would still be dominated by the similarity of high-frequency processes.*

*Accordingly, we assumed that the influence of the annual cycle on the relative distribution (i.e. the relative change of a metric in space around a TG) on RMS and correlation would be minor. However, we did not quantify the contribution of the annual cycle to those metrics nor evaluated our assumption. Hence, in response to the reviewer we repeated the analyses by first de-trending and de-seasoning the data (before computation of the statistics.)*

*In the following plot we compare the impact of de-seasoning on the metrics RMS, correlation and residual annual cycle. In this plot, we also use weighted RMS or standard-deviations of the trend differences (SAT-TG minus GPS) as suggested in a subsequent comment:*

[Figure]

*Figure R1: Comparison of statistics when the data was de-seasoned (left) or not (right) before matching. First row shows the unweighted RMS, the second row shows the weighted RMS of trend differences, the third row shows the un-weighted standard deviation and the last column shows the weighted standard deviation. The red bar marks the level range, which had been identified as the global optimum based on the un-weighted RMS.*

*In the original version (detrended and not de-seasoned) the metrics RMS and correlation were shown to provide very similar results (in terms of accuracy and uncertainties). When we de-season the data (prior to computation of the metrics), we do not find significant improvements (in accuracy) or effects on the use of correlations or RMS, as we also concluded in the study. To assess the influence of de-seasoning on uncertainties, we reconstructed Figure 4, now with results based on metrics derived from detrended and de-seasoned time series:*

**A. Original results**

[Figure]

**B. Results after de-seasoning and detrending**

[Figure]

*Figure R2: Shown are A) statistics when data was only detrended but not de-seasoned; B) statistics based on detrended and de-seasoned data. Figure captions are as in Figure 5: Performance of VLMSAT-TG trend estimates for ALES-GESLA-ZOI. a) RMSΔVLM for different relative thresholds (step size 2.5%) and different selection criteria: RMS SAT-TG (blue), correlation (red) and residual annual cycle (AC, green); c) same as (a) but for median uncertainties. b) Distribution of best performing relative thresholds for individual stations. The local optimal threshold is defined at the minimum of the absolute difference of VLMSAT-TG and GNSS trends. d) Boxplot shows the distribution of the mean distances to coast for the individual optimum ZOI's as denoted in b). The distances refer to the distributions within the 0-25%, 25-50%, etc. levels, respectively.*

*Hence, in our application de-seasoning of the data does not significantly alter the choice of the statistics (on which the ZOI is based on). To better justify our choice, we added the following lines in the manuscript L270:*

*'The statistics are based on de-trended data. Thus, all the metrics may be influenced by the similarity of the annual cycle. However, by repeating this analysis using de-trended and de-seasoned data (not shown), no significant differences were identified.'*

**L271-273: part2: In addition the authors fit the seasonal variation together with the linear trend in the SAT-TG series, i.e., the residual seasonal variation may play a minor role in the estimated VLM and its uncertainty. I'm not sure if this is what the authors intended. It may not have a significant impact on the selected ZOI areas, but the authors would be at least using more independent criteria.**

*To assess the trend components of the SAT-TG VLM time series we followed the standard approach to estimate also the seasonality as done in previous studies (e.g. Wöppelmann and Marcos, 2016). In case that the residual annual cycle is caused by differences in observed SLA variability, we still parameterize such variations, since we are interested in the long-term changes and the uncertainties arising from long-term variations. The annual cycle signal, which we assume to be constant over time, should not contribute to uncertainties associated with changes on times scales longer than one year.. Another factor, which however only has a minor impact, is that at some locations the SAT-TG series might even contain annual signals that exist due to actual seasonality of the local VLM, but not due to the residual annual cycle between the SAT-TG measurements. Such signal should then be modelled; otherwise, it would increase the trend uncertainties and it would be inconsistent with the GNSS trend estimates (where seasonality was also taken into account).*

**L367: The authors take the GNSS VLM as ground truth, and that is fine, but are the formal VLM errors similar among the TG and GNSS stations, respectively? Formal errors can provide valuable information for the VLM validation and this should be accounted for when assessing the VLM differences (WRMS instead of RMS for instance).**

*Formal errors are still much lower for the GNSS VLM estimates than for the SAT-TG estimates. To take these into account, we re-computed the weighted RMS as well as the weighted STD of the trend deviations as follows (also as a response to one of the subsequent comments):*

$$RMS_{weighted} = \sqrt{\sum_{i=0}^{n} w_i (GPS_i - SATTG_i)^2}$$

$$STD_{weighted} = \sqrt{\sum_{i=0}^{n} w_i ((\overline{GPS - SATTG}) - (GPS_i - SATTG_i))^2}$$

*With weights* $w_i = \dfrac{\sqrt{(GPSuncertainty_i^2 + SATTGuncertainty_i^2)^{-1}}}{\sum_{i=0}^{n} \sqrt{(GPSuncertainty_i^2 + SATTGuncertainty_i^2)^{-1}}}$

*Table 1: Comparison of SAT-TG minus GPS trends: first column: RMS not weighted as in manuscript, 2nd: weighted RMS and 3rd: weighted standard deviation.*

|  | RMS-normal | RMS-weighted | STD-weighted |
|---|---|---|---|
| ALES_GESLA_250km | 1.51 | 1.47 | 1.39 |
| ALES_PSMSL_250km | 1.68 | 1.57 | 1.46 |
| ALES_GESLA_ZOI | 1.28 | 1.37 | 1.19 |
| AVISO_PSMSL_250km | 1.50 | 1.48 | 1.32 |

*Using a weighted RMS most strongly improves the ALES_PSMSL_250km configuration, but it has a smaller effect on the other data sets. This is an interesting finding, which shows that lower formal uncertainties are not in every case associated with more accurate trend estimates. Such de-coupling of accuracies and uncertainties was also addressed in the discussion and points towards other undetected error sources, which limit the comparability of SAT-TG and GNSS.*

*We added the formulation of the weighted RMS in section 3.4 and associated results in table 2.*

*A weighted STD improves all of the datasets, because here the mean bias of the differences does not have an impact on the performances anymore.*

*We decide not to add the (weighted) STD in the table, but add a more thorough discussion on causes of trend biases in the section 5.2. Systematic errors (for more details please refer to next comment).*

**L377-380: the differences between the SLA trends in ALES and AVISO are quite significant. From Table 2 it appears that the median SLA value from AVISO is 1 mm/year higher than that from ALES (also seen in Fig. 4). Is this correct? If so, how would you explain this difference?**

*We agree that there is a strong difference between the median (or as we call it the bias) of the two dataset combinations. Also comparing the bias of AVISO-PSMSL with the result from Wöppelmann and Marcos 2016 we obtain a much larger value. We briefly addressed this issue by pointing out the differences in the settings of the different studies (250km range instead of 1° average, other time periods and TGs (numbers and locations)). Also in response to the second reviewer, we integrated the discussion of the impact of possible mission drifts, which could generate systematic trend biases. We added some further lines in the introduction as well as in the results section:*

*Methods (we add more information on our cross-calibration analysis in L170):*

*'To reduce radial errors in the different missions, the tailored coastal altimetry product is cross-calibrated using the global multi-mission crossover analysis (MMXO)  (Bosch and Savcenko, 2007; Bosch et al., 2014). The MMXO minimizes a large set of globally distributed single- and dual sea surface height crossover differences by least-squares adjustment. The estimated radial errors are used to correct each individual sea surface height measurement. In this way, we not only reduce orbit inconsistencies, but also those originating from the range and from applied corrections. Since we estimate a radial correction for each observation, we minimize intermission drift differences as well as regionally correlated errors. Note that this approach is a relative calibration and provides range bias corrections with respect to NASA/CNES reference missions. Any remaining absolute drift of these reference missions (with respect to TGs) still influence the drift of the whole altimeter solution.'*

*In the results section, we add following lines to the paragraph from L385:*

*'For both combinations, the  median of the VLM  differences (ALES-PSMSL-250km -0.87 mm/year AVISO-PSMSL-250km: 0.56 mm/year)  deviates from values shown in previous studies [WM16: -0.25 mm/year and, Kleinherenbrink et al. (2018): -0.06 mm/year]. In contrast to these previous estimates, we use different spatial selection scales of SLAs, smaller numbers of TG-GNSS pairs and deviating record lengths, which impedes a direct comparison. Moreover, the altimetry datasets might be affected by instrumental drifts. In this respect, differences among the datasets may be caused not only by different techniques applied to reduce intermission biases (e.g., the MMXO approach for ALES), but also by different missions incorporated in the records. Note that in contrast to ALES, AVISO contains TOPEX, which has also been shown to be affected by a strong drift (Watson, 2015).'*

*We added some additional statistical analysis in the supplemental material (from L581):*

*'To gain a better understanding why the VLM SAT-TG and VLM GNSS difference distributions are significantly biased, we create a Monte-Carlo experiment to check the H0 hypothesis: 'the median of the distribution is not significantly different from zero' (with alpha=0.025). Therefore, we generated a bootstrapped distribution of random medians, which are derived from 20,000 individual sub-sets of size 52 (the number of TG of our dataset), which are randomly drawn from normally distributed values with a standard deviation of 1.5 mm/year (according to the RMS of AVISO-PSMSL-250km) and zero mean.*

*Figure 1 shows that the biases of the datasets ALES-PSMSL-250km and AVISO-PSMSL-250km exceed the 2.5 and 97.5 percentiles of the sampled distribution (average of absolute bounds: 0.512 mm/year). This means that in less than 5 out of 100 cases, we would obtain such biases by chance, which supports the significance of these biases. We highlight that this is a purely statistical analysis, which cannot account for any of the errors from corrections, adjustments, drifts etc. introduced in the altimeter and GNSS.'*

[Figure]

*'**Figure B1:** Histogram of median values of randomly sampled sub-sets. The sub-sets consists of 52 samples (according to the number of TGs in AVISO-PSMSL-250km) and are randomly drawn from normal distributed values with zero mean and a standard deviation of 1.5 mm/year (according to the RMS of AVISO-PSMSL-250km). Dashed lines mark the 2.5 and 97.5 percentiles of the distribution.'*

*Given these results, we dedicated another sub-section to trend biases and systematic errors in the discussion (section systematic errors, after L524):*

*'5.2 Systematic errors*

*VLM estimates from different datasets (e.g. AVISO-PSMSL-250km and ALES-GESLA-ZOI) are biased compared to trends inferred from GNSS observations. Based on Monte-Carlo simulations (see appendix Figure B1) we argue that these biases are significant for most of the dataset combinations*

*(ALES-PSMSL-250km and AVISO-PSMSL-250km, ALES-GESLA-ZOI). In the following, potential sources for these biases will be discussed.*

*Next to the record-length (see section 5.1), systematic errors critically affect the accuracy of the SAT-TG technique and can have strong systematic effects on the trend differences. Limiting factors for VLM determination from both SAT-TG and GNSS observations are the accuracy and uncertainty of origin and scale of the reference frame (see WM16, Collilieux and Woppelmann (2009), Santamaría-Gómez et al. (2012)), which cannot be realized yet at the required accuracy level (Bloßfeld et al, 2018; Seitz et al., submitted).*

*Moreover, as mentioned before, the intermission calibration applied for ALES (MMXO) reduces intermission biases, but does not feature a calibration against TG. The median bias identified for ALES-GESLA-ZOI could be affected by a drift of the mission used as reference. In contrast, the AVISO dataset does not include time-dependent intermission biases and might therefore be additionally influenced by systematic effects of e.g. Envisat or Sentinel-3a (Dettmering and Schwatke, 2019).*

*Next to altimeter bias drift, non-linear VLM from contemporary mass redistribution (CMR) changes were shown to cause differences between VLM_SAT-TG and VLM_GPS, due to the different time periods covered (e.g. Kleinherenbrink et al. (2018)). Using GRACE (Gravity Recovery and Climate Experiment) observations, Frederikse et al. (2019) demonstrated that associated deformations can cause VLM trends in the order of 1 mm/year. Therefore, they introduced a new method to reduce VLM_GPS by GIA and CMR signals to minimize their associated induced extrapolation biases. Kleinherenbrink et al. (2018) incorporated non-linear VLM from CMR to assess the corresponding trend differences between VLM_SAT-TG and VLM_GPS. They exposed that VLM_SAT-TG estimates are lower than VLM_GPS in many parts of North America and Europe and higher in subtropical/tropical regions as well as Australia and New Zealand (refer to Figure 9 in Kleinherenbrink et al. (2018)). Because northerly regions, for instance, are affected by stronger recent uplift, GNSS observations which cover shorter and more recent time spans than satellite altimetry detect more positive trends. For a set of 155 TG-GNSS pairs, integration of these signals slightly reduces the median bias from -0.14 mm/year to -0.07 mm/year, but had no significant effect on RMS. Given that most of the TG-GNSS stations used in this study are located in Europe, North America and Australia, CMR might as well alleviate the negative trend bias of ALES-GESLA-ZOI. Therefore, extending the validation platform, not only by using other homogeneous GNSS observations, but also GRACE and GIA estimates would support the identification and mitigation of such systematic errors.'*

*Bloßfeld M., Angermann D., Seitz M.: DGFI-TUM analysis and scale investigations of the latest Terrestrial Reference Frame realizations. In: (Eds.), International Association of Geodesy Symposia, 10.1007/1345_2018_47, 2018*

*Dettmering D., Schwatke C.: Multi-Mission Cross-Calibration of Satellite Altimeters - Systematic Differences between Sentinel-3A and Jason-3. International Association of Geodesy Symposia, 10.1007/1345_2019_58, 2019*

*Seitz M., M. Bloßfeld, D Angermann, M. Gerstl, F. Seitz: DTRF2014: The first secular ITRS realization considering non-tidal station loading. Journal of Geodesy, submitted.*

**L386: "absolute median bias of trend differences" can be confused with the median of absolute VLM differences. I suggest changing this by "median of the VLM differences" or similar here and elsewhere.**

*Corrected to: 'For both combinations the  median **of the VLM**  differences (ALES-PSMSL-250km:  -0.87 mm/year AVISO-PSMSL-250km: 0.56 mm/year)  deviates from*

*values shown in previous studies [WM16: -0.25 mm/year and, Kleinherenbrink et al. (2018): -0.06 mm/year].'*

**L400-401: Some comments on results shown in Table 2: the unweighted RMS from GESLA is smaller than that from PSMSL, but this is probably because the median value is closer to zero. The weighted standard deviation or any other measure of dispersion (interquartile range) that does not include the mean/median value would be more appropriate here. Also the formal VLM rate uncertainties are higher with GESLA than with PSMSL, even with a spectral index slightly closer to zero. This means the noise (especially the power-law variance) of the residual series (trend and seasonal variations removed) in the GESLA VLM series is larger than that from PSMSL or that the GESLA series are significantly shorter or less complete. In that case, the choice of using GESLA instead of PSMSL would need better argumentation. There may be also a TG trend bias between GESLA and PSMSL of around 0.3 mm/year. This is probably not significant, but it may be worth discussing.**

*In a previous comment, we added the statistics weighted RMS as well as weighted STD, which confirm that ALES-GESLA-250km still outperforms ALES-PSMSL-250km in terms of accuracy. As mentioned by the reviewer, larger trend uncertainties for the GESLA configuration can be a result of larger power-law variance. We found that, the median driving noise of ALES_GESLA_250km is by 5% larger than for ALES_PSMSL_250km. Thus, we further discuss such potential causes of the trend uncertainty differences by adding to L402:*

*'Compared to ALES-PSMSL-250km, we find increased trend uncertainties for ALES-GESLA-250km, which can be partially explained by higher power-law variance of this GESLA -based configuration. '*

*! Due to the update of Envisat data (see first response), the mentioned 0.3 mm/year difference of trend biases (ALES-GESLA-250km vs. ALES-PSMSL-250km) increased to 0.48 mm/year !*

*Relating to our previous response, showing the probability of occurrence of a median, we argue that for such a sample size a trend bias of 0.48 mm/year is not significant. Please also refer here to the discussion of the impact of systematic errors on trend biases (section 5.2). The general question raised by the reviewer, which requests for a better justification of the use of one TG dataset over another needs to be better assessed. Therefore, we add following lines to the previous corrections (L402):*

*'… variance of this GESLA -based configuration. Although trend uncertainties are higher for the ALES-GESLA-250km configuration, we choose this set-up to investigate the impact of the ZOI. This dataset provides better results concerning trend accuracy (weighted or unweighted RMS) and has a lower median bias. Moreover, using the high-frequency data, we are able to couple SAT and TG observations at much higher temporal resolution than it would be the case when using monthly PSMSL data.  Therefore, the ALES-GESLA coupling is further developed based on a better definition of the ZOI in the next section. '*

**L421-422: the power-law variance may have also changed, maybe producing a significant improvement of the VLM formal errors.**

*We added: 'stems from the reduction of the power law and white noise amplitudes'*

**L429-436: The discussion in this paragraph is not very clear and would require improvements. We only need a single SLA series to estimate VLM from SAT-TG. This SLA series can be obtained using different strategies as the authors have discussed: spatial averaging/filtering of SLA data, the single most correlated SLA series, the single**

**closest SLA series, etc. The more similar the selected SLA series is to the TG series, i.e., the smaller the SAT-TG differences (again excluding the seasonal variations that are captured by the model fitted to the SAT-TG series), the more precise VLM will be obtained in terms of formal error. This is a metric very easy to interpret. Note that interpreting how the SAT-TG VLM values compare to the GNSS VLM is much more complex and is, in general, not a strong criterion given the large differences between both. The smaller quantity of averaged SLA should not be blamed if they represent increasingly consistent SLA series with respect to the TG series. A different explanation for the bad results with >80% thresholds could be that the RMS metric is not telling us whether the SAT and TG series are more similar, especially if the seasonal signals were included in the RMS as per my comments above. In addition, the RMS alone is not directly tied to the autocorrelation of the SAT-TG series (i.e., the spectral index), which is another important metric to assess the consistency of the SLA and TG series.**

*In this paragraph (L429-436), we discuss why the RMS (of SAT-TG and GNSS trend differences, i.e. the accuracy of trend estimates) increases when we select very highly comparable data, or smaller subsets of altimetry data. Overall, we still argue that at very high levels (which can also mean less selected tracks) a mere decrease in sample size of the time series is the major reason for decreased accuracies of the SAT-TG trends. As an example, a 95% level-ZOI selection (based on RMS) would only hold 80% of the samples (i.e. number of monthly averaged observations) which we would obtain at the 80% level-ZOI selection. Considering the subsequent analysis in the discussion (e.g. dependence of accuracies on the length of the covered time period) this is in our understanding the most obvious explanation for decreased accuracies, when we strongly decrease the sampling density at high levels of comparability.*

*In general, we fully agree that there is a large range of error sources which influence the comparability of SAT-TG and GNSS trends. However, when we only adjust the amount of selected SAT observations, we keep much of those error sources constant (mission drift biases, nonlinear VLM, some of the errors in applied corrections …). Thus, because we reduce the number of samples e.g. at a 95% level, compared to a 80% level we came to this conclusion.*

*Therefore, to better clarify our explanations we modify the paragraph as follows (from L429):*

*'RMS_VLM and trend uncertainties level off at very high thresholds and ultimately increase when only 5% of the data is used (Figure 5a and 5c). We argue that this is mainly related to a decrease in sampling-density of the time series included in the selection: At the 95$^{th}$ percentile, the median sample size (i.e. number of monthly averages in a time series) is 20% smaller that the sample size at the 80th percentile. Robust trend estimates require a minimum of samples, hence, using a reduced number of along-track data time series, even when they show a maximum degree of comparability, yields on a global average decreased trend accuracies (RMS_VLM).  We thus argue that the optimum threshold identified at about the 80th percentile (of the data sorted by RMS) represents a compromise between data-comparability, as well as sampling-density of altimetry data. We emphasize that there are numerous factors, other than the time period covered, which may contribute to a lack of comparability of SAT-TG and GNSS trends. We further elaborate those in the subsequent discussion section 5.'*

*The reviewer argues that, because we model the annual cycle this might be one reason why our metric, the RMS of SAT-TG differences, is not telling us whether SAT and TGs are more similar to each other and thus inadequately expresses deviations between SAT and TG time series. We note, however, that this metric indeed also captures differences in the annual cycle between both time series (see previous discussion, because the RMS is computed based on the differences of the*

*detrended but not de-seasoned time series). We then use this metric to confine the ZOI, to average data from which we again compute SAT-TG time series. Concerning the spectral index, we agree that the RMS is not directly related to this metric.*

**L466-478: the discussion here is interesting, but an important point should be stated more clearly. The optimal ZOI in Fig. 5 a&c was retained by assessing the consistency between SAT and TG series. On the other hand, the local optimal ZOIs in Fig. 5b are defined by comparing SAT-TG and GNSS VLM estimates. It is therefore not surprising that different optimal local ZOIs are obtained and that there is not an optimal threshold that fits well all sites. In addition, imposing the GNSS VLM as a criterion to the computation of the SAT-TG VLM would remove its independent nature.**

*We change this paragraph to better clarify the meaning of the results shown in Figure 5 as well as the aim of our interpretations of Figure 5 (from line 460-478):*

*'The results presented in Figure 5 and Table 2 denote  metrics and performances derived from the global TG-GNSS dataset for ALES-GESLA-ZOI  and support an optimal threshold at 20%. It is however unclear, whether the described optimum threshold for this 'global' selection also reflects the best choice at every  coastal site considered. Therefore, we investigate at which relative levels individual VLM SAT-TG and VLM GNSS trends estimates yield the smallest absolute deviations. Postulating that the actual VLM at the TG location is linear and perfectly detected by the GNSS station, these thresholds denote the 'local' optimal levels. With this analysis, we aim to better understand the spread of individual optimal ZOIs and what would be the best theoretically achievable RMS_ΔVLM. This analysis also provides a basis to motivate future investigations, in particular to identify systematic factors, which may lead to local different extents of the ZOI and to improve the accuracy of trend estimates.*

*Figure 5b displays the distribution of local optimal thresholds for TG-GNSS stations for the ALES-GESLA-ZOI dataset. Note that these estimates are not independent as they are based on prior knowledge of the ground truth VLM from GNSS. … An associated ideal selection of trends, based on optimal individual levels shown in Figure 5d would largely reduce to RMS_VLM to  0.89 mm/year. We emphasize that this constitutes the best RMS, which could theoretically be achieved with our dataset combination, if all of the local optimal levels could be systematically explained. This demonstrates that, albeit there might be room for minor improvements, there is still a strong limitation remaining in bringing the RMS below 1 mm/year.*

**L487-488: I guess you derived the 3 to 4 inflation factor from the color range in Fig. 6, but Table 2 actually shows that using different SLA data does increase VLM uncertainties by less than 10% (comparing ALES-PSMSL to AVISO-PSMSL).**

*Exactly, we better specify these lines (L487-488):*

*These examples show, that at individual locations  the use of less comparable SLAs can increase uncertainty by a factor of three to four.*

---

## Author Comment (AC2) · 27 Aug 2020

We thank Christopher Watson for his very helpful and comprehensive comments and his profound corrections. His remarks pointed out several vital aspects, which were so far missing or insufficiently covered in the manuscript. In response to his comments, we added more precise explanations, specified technical details and discussed the results much more thoroughly. Inclusion of these aspects greatly helps to enhance the manuscript and makes it more comprehensible and complete.

On behalf of all authors, Julius Oelsmann

[Figure]

Please also note the supplement to this comment:
https://os.copernicus.org/preprints/os-2020-29/os-2020-29-AC2-supplement.pdf

[Figure]

**Supplement:**

Review response

To Christopher Watson, August, 26 2020

*We thank Christopher Watson for his very helpful and comprehensive comments and his profound corrections. His remarks pointed out several vital aspects, which were so far missing or insufficiently covered in the manuscript. In response to his comments, we added more precise explanations, specified technical details and discussed the results much more thoroughly. Inclusion of these aspects greatly helps to enhance the manuscript and makes it more comprehensible and complete.*

*General note: We changed the Envisat mission data version from V2.1 to V3 (for the ALES altimetry data). This influences all associated dataset combination (ALES-PSMSL-250km, ALES-GESLA-250km and ALES-GESLA-ZOI). Because the statistics are not significantly altered, the key messages of the study remain. All statistics and plots are updated accordingly (i.e. Figure 1,2,4,5,6,7 and table 2).*

*We use italic formatting to answer the comments. Existing text is marked in blue and changes in the text are highlighted in red. All line numbers refer to the originally submitted version.*

**This manuscript by Oelsmann et al provides a contribution to the improved determination of vertical land motion (VLM) at tide gauge locations using satellite altimetry. The work advocates for a refined approach to selecting valid altimeter data in a variable 'zone of influence' around a specific tide gauge to improve the accuracy and precision of VLM estimates. The team make incremental advances using a coastally retracked altimeter dataset and tide gauge data with various temporal resolutions. The focus of the paper has a rich history in the literature and this technical contribution provides a worthy contribution to progressing the method and ultimately achieving improved VLM estimates around the coast. The demand for improved VLM is clear – VLM is a vexing issue with manifold impacts across the scientific and the broader community. The manuscript is well presented, well crafted and generally presents a compelling and comprehensive case worthy of publication in this outlet. Noting my support for this manuscript, I have a few substantive issues that I feel would benefit from consideration and discussion/revision in the paper. I present these issues below, followed by a list of technical clarifications and minor suggestions/corrections that may benefit the manuscript.**

**Main Issues:**
**Central to the manuscript is the comparison of various altimeter minus tide gauge differences against a single 'ground truth' GPS solution. Significant differences in the solutions derived from different variants (e.g. Fig 4b) are evident, highlighting the sensitivity of the problem. This suggests that fundamental issues remain to be understood in one or more of the constituent components used (i.e. any one or all of ALT/TG/GPS). Relating to this point and notwithstanding my remarks above regarding the relevance of this work, my sense from reading the manuscript is that the authors pay too little attention to discussing a few key issues that I argue should otherwise appear in a paper such as this:**

**1) Despite the technique itself originally emerging at the time investigators were using tide gauges (and thus VLM) to validate the stability of the altimeter measurement system, the authors do not seem to acknowledge the potential for regionally correlated error within the altimeter – brief mention of the importance of this point in the heritage (and current use) of the technique is also surprisingly lacking. Rather, the authors make the bold assumption of 'no instrumental drifts' (pg 3, ln 75) as the sole**

**mention of possible systematic error in the altimeter record. I note the authors exclude TOPEX/Poseidon from the coastally retracked dataset given waveforms are presently unavailable, however, it is included in the AVISO product. The well-known issues associated with TOPEX (brought to light using this technique and later confirmed) put**
**the issue into focus. Further, inspection of solely orbit differences in a regional context places even greater emphasis on the issue (e.g. Couhert et al, ASR (2015) who report on the significant challenge of achieving 1 mm/yr orbit stability at regional scales). The use of a consistent orbit across all missions in this current paper (which I assume are the GFZ Ver13 SLCCI orbits?), will assist to mitigate this effect but it is likely that mission-specific, spatially-correlated 'whole-of-system' drift will remain in the residuals and undoubtedly make at least a contribution to the differences observed in the current paper. I contend that 'no instrumental drifts' is somewhat of a misnomer in the context of whole-of-system altimetry at regional scales. Not mentioning these broad issues nor the heritage of the ALT-TG technique in this regard would be disappointing for a paper such as this.**

*We highly appreciate this comment, because it holds valuable and necessary information concerning fundamental underlying errors of the ALT-TG technique, which we had not covered so far in the pa-per. We make several adjustments in the text to integrate these issues:*

**_Systematic errors (regionally correlated errors, altimeter drifts):_**

*Introduction (we add short mentioning of systematic errors) in L75:*

[revised manuscript text omitted]

*Bloßfeld M., Angermann D., Seitz M.: DGFI-TUM analysis and scale investigations of the latest Terrestrial Reference Frame realizations. In: (Eds.), International Association of Geodesy Symposia, 10.1007/1345_2018_47, 2018*

*Dettmering D., Schwatke C.: Multi-Mission Cross-Calibration of Satellite Altimeters - Systematic Differences between Sentinel-3A and Jason-3. International Association of Geodesy Symposia, 10.1007/1345_2019_58, 2019*

*Seitz M., M. Bloßfeld, D Angermann, M. Gerstl, F. Seitz: DTRF2014: The first secular ITRS realization considering non-tidal station loading. Journal of Geodesy, submitted.*

**Orbits:**

*Because we use ITRF2008 (as our GNSS solutions are based on this system), we don't use GFZ Ver13 SLCCI orbits that are in ITRF2014. For Jason3 and Saral we use the CNES GDR-E orbits and for ERS2 the GFZ VER11 orbits. The original orbits based on the GDR data are used for Envisat, Jason1 and Jason2. The reference system is consistent for all orbits (ITRF2008). Small differences due to smaller processing differences are accounted for in MMXO.*

*We specify the orbits further in lines (L156):*

*'For all missions satellite orbits in ITRF2008 are used, mostly processed by CNES (GDR-E). For ERS-2, GFZ VER11 orbits are applied.'*

**2) A second issue of less significance relates to the treatment of the tides in the high frequency tide gauge data. The authors attenuate the tide with a 40-hour loess filter (pg 8, ln 217). Clarification and further defence of the filter strategy would be beneficial here (e.g. both are likely small, but I wonder what % of \*non\*-tidal variance is attenuated by the filter, and what % of \*tidal\* variance at longer periods is retained (but removed from the altimeter with its model-based correction)). The comment regarding the temporal resolution of the DAC used for the high rate TG data felt ambiguous here (ln _219), and should be clarified with respect to consistency with the filtering approach. The magnitude of the difference in tides between (coastal) altimetry that uses a global model and at the tide gauge is an important issue that warrants at least a mention in the discussion.**

*Removal of TG tides: To remove the tidal signal from high frequency TGs, we followed the approach by Saraceno (2008), who used the same 40h-loess filter. We did not use a global tide model to correct for tides, because the quality of the modelled signals in shallow and coastal waters (i.e. near the Tide gauges) deteriorates. As an example, Piccioni (2018) showed that, regardless of the improvements of using dedicated coastal altimetry, the variance of residual tidal components (after subtracting tidal signal based on FES2014) can still remain in the order of several cm.*

*We add more information on the impact of our filtering approach on several tidal constituents in a qualitative way. For our considered GESLA TG dataset (consisting of 58 stations) the tidal filtering reduces the median standard deviation of the data by 27.5%. Note that such a reduction is strongly location-dependent. The following plots illustrate the reduction in tidal variance at different periods for selected TGs:*

*A)*

[Figure]

[Figure]

B)

[Figure]

*Figure R1: Time series (upper rows) and spectral densities (lower rows) for two different TG stations (A) France and B) Brazil). The original time series is shown in blue and the filtered one in orange. Spectral density estimates are obtained from using Welch's method and are based on time series with at least 2 years of data. Data gaps in the time series are filled by linear interpolation, with a maximum allowed data gap of 10 hours.*

*The plots show that for time-scales up to ~1 day spectral power is reduced by about two orders of magnitude. For time scales larger than 2.5 days, the filter does not reduce any of the longer periods. At the tide gauge in France (Figure R1 A) the long period tidal components Mm and Mf have amplitudes of 0.889 cm and 1.143 cm, respectively (based on the TICON dataset, Piccioni et al., 2019). At the second TG in Brazil (Figure R1 B) the amplitudes are 0.475 cm (Mm) and 1.288 cm (Mf). The plot below illustrates the amplitudes of these long period tidal components. At Mf frequency, tidal corrections applied to TGs or altimetry can deviate from each other with an amplitude up to ~2.5 cm.*

[Figure]

*Figure R2: Modelled (FES2014) SSH amplitude of two fortnightly and monthly tidal constituents: Mf (left, period 13.66 days) and Mm (right, period 27.55 days)*

*Piccioni, G, Dettmering, D, Bosch, W, Seitz, F. TICON: TIdal CONstants based on GESLA sea-level records from globally located tide gauges. Geosci. Data J. 2019; 6: 97– 104.*
*https://doi.org/10.1002/gdj3.72*

*We are grateful to the reviewer for pointing this out and we reckon that we could further improve the comparability between TG and altimetry by accounting for longer period tides, for example performing a tidal harmonic analysis directly on each TG record. We keep this as an interesting development for our further studies and for now we clearly state in the text the caveat of our current methodology. Therefore, we added in L218:*

*'This filtering approach most effectively reduces tidal variance at periods lower than ~2 days (e.g. reduction by more than two orders of magnitudes at daily periods). However, tidal variability at periods larger than 2 days is not significantly attenuated by the filter. Therefore, one caveat of this approach is that there remains residual tidal variance at longer periods between TGs and altimetry, given that the latter features a model-based adjustment for longer tides. We do, however, not apply the same tidal model to the TGs, due to known issues related to decreased model performance in shallow water (Piccioni et al., 2018). '*

*Piccioni, G., Dettmering, D., Passaro, M., Schwatke, C., Bosch, W., and Seitz, F.: Coastal Improvements for Tide Models: The Impact of ALES Retracker, Remote Sensing, 10, 700, https://doi.org/10.3390/rs10050700, 2018.*

*Regarding the 6h-resolution of DAC (and interpolation on hourly TG data): As in Saraceno et al. (2008) we also use a 6h-resolution DAC correction and apply it after filtering the TG data. As the best compromise, we decided to interpolate (cubic) this correction onto the hourly time step of the TGs (L220). Overall, this approach ensures consistency between the DAC correction for TGs and altimetry, which requires the same temporal resolution as subsequently TG and SAT time series and therefore, the DAC correction are subtracted from each other.*

**3) Finally, the improvement observed between along-track altimetry v gridded altimetry is presented in the context of 'coastal altimetry' driving the improvement. I'm very pleased to see these results however I feel that conclusion needs some further defence. From my understanding of Fig 5d, the lower quartile of the data commences at just under 30 km from the coast (and increases from there). Discussion of Fig 5b in the text (pg 22, ln 470) refers to a much smaller subset in which the mean distance was 33 km. Given the benefit of retracking is 'most pronounced in the last 20 km' the authors should further evidence what % of their data is within this threshold and thus further elucidate whether it is simply the influence of along-track data versus its retracked equivalent. To be clear, I am not asking for the former to investigated in the same way – the benefits of the retracking in general terms are clear – I just wish to ensure the conclusions are appropriately elucidated.**

*As the reviewer noted, the mean distance of 33 km referred to a smaller sub-set comprising all ZOIs within the 0.8 – 0.975 levels. To clarify how much of the data is near the coast at the global optimum, we now refer to the 80$^{th}$ level (based on the correlation metric.)*

*We adjust the text as follows:*

*At the global optimum itself (0.8, based on correlations), the median distance to coast (of all SLA measurements in a ZOI) is 39.4 km. 25 % of the altimeter observations are within a range of 20 km to the coast, i.e. the region with the most pronounced coastal advancements of the along-track dataset (Passaro et al., 2015).*

**Technical Issues:**

**It is unclear throughout if the RMS statistics computed against the GPS time series take into account the GPS uncertainties? I expected to see WRMS mentioned rather than RMS.**

*The RMS statistics as used in the study is the unweighted RMS. To account for GPS and SAT-TG formal uncertainties we added the weighted RMS in table 2 (pg 17.) and adjusted paragraph 3.4. Comparing the weighted and un-weighted RMS, we find that not in all cases accounting for formal uncertainties is associated with better accuracy.*

*Because the weighted RMS supports the use of ALES-GESLA over ALES-PSMSL as well, we add (L402):*
*' … Although trend uncertainties are higher for the ALES-GESLA-250km configuration, we choose this set-up to investigate the impact of the ZOI. This dataset not only provides better results concerning trend accuracy (weighted or unweighted RMS) and has a lower median bias. …'*

**In section 2.1, I assume the GFZ Ver 13 SLCCI orbits are used homogeneously for all missions, but this is only inferred. I suggest adding detail to Table 1 for clarity and citing the relevant Rudenko et al paper.**

*Please refer to the answer to issue #1 (more details on the applied orbits)*

**I was pleased to see the data has been cross calibrated using the MMXO approach (pg 6, ln 165). Some additional detail here would be beneficial: I assume this provides a location-specific and mission-specific radial bias, computed using the \*identical\* dataset (Table 1). Some further detail on this would be useful here. Further, some comment on the relative trends observed (and applied?) between the Jason-series and ERS/Envisat missions (using the MMXO approach) is warranted here.**

*As the reviewer noted, the approach uses a very large set of location-specific, single- and dual-satellite crossover differences to reduce the radial errors of the missions. Therefore, the technique enables us to minimize the geographically correlated mean errors. This approach is applied to the \*identical\* dataset (based on all corrections and adjustments as listed in Table 1).*

*Please refer to the additional information given as a response to the first comment or changes from L170.*

*To illustrate the magnitude of the MMXO correction, we show the estimated relative range bias (10 day means) for Envisat V3 w.r.t. Jason-1 (Fig. R3) indicating the temporal systematics which are accounted for by the MMXO. Fig. R4 shows the geographically correlated mean errors (offset subtracted) between Envisat and Jason-1. Both effects are removed from the Envisat data by applying the estimated radial corrections for each measurement.*

[Figure]

*Figure R3: Envisat relative range bias per 10-day cycle (w.r.t. Jason-1) [mm]*

[Figure]

*Figure R4: Geographically correlated mean errors of Envisat w.r.t. Jason-1 [mm]*

**Regarding the pole tide in Table 1, could the authors confirm they only apply the solid
Earth component of the pole tide (noting the pole tide that is present in the GDR combines
the ocean plus solid earth component, hence needs to be multiplied by h2/(1+k2).**

*We are using our own software to correct for solid earth and pole tides following the IERS conventions
(2010) instead of relying to the corrections provided in GDR. Thus, we can confirm that point.*

**Regarding the ocean tide in Table 1, it might be worth adding a paragraph to further
discuss the long-period tides that are/are not considered in the altimeter data. The
current treatment of the tide in the high frequency dataset at least should retain any long period
constituents – confirmation that is the case with the altimetry would also be
beneficial.**

*Please refer to the answer of the second comment.*

**Regarding the cubic interpolation to the hourly tide gauge data, I'm assuming you don't use this across outages larger than some threshold? Worth clarifying. In Figure 2, this may be more beneficial to show the data in panel 3 over the data in panel 2. Then, in the now vacant third panel, show the same but with the annual and semi-annual terms estimated and removed. Note the caption reports "SAT minus TG" but the y-axes report "TG-ALT".**

*Concerning the combination of hourly TG data and altimetry data, we only match values (or interpolate values) with data gaps of maximal 3 hours. This is added in the text in L235: '' … by cubic interpolation of the latter and a maximum allowed time-lag of 3 hours between the measurements.'*

*Figure 2: We changed the Figure and its caption accordingly and corrected the axes descriptions.*

**In Figure 4, I found Fig4b and d difficult to interpret. I feel they need to be introduced earlier and/or more comprehensively explained as I felt it took me a few reads to get the point.**

*We assume that the reviewer refers to Figure 5 (Given that there would not be differences between Figure 4a-d). We added more explanations to better introduce the figure and to better understand the purpose of why we performed this analysis. We also clarified our interpretations of Figure 5b:*

*We changed this paragraph (in which Figure 5 is explained) (from line 460-478):*

*'The results presented in Figure 5 and Table 2 denote  metrics and performances derived from the global TG-GNSS dataset for ALES-GESLA-ZOI  and support an optimal threshold at 20%. It remains to be investigated,  whether the described optimum global threshold  also reflects the best choice at every  coastal site considered. Therefore, we investigate at which relative levels individual VLM SAT-TG and VLM GNSS trends estimates yield the smallest absolute deviations. Postulating that the actual VLM at the TG location is linear and perfectly detected by the GNSS station, these thresholds denote the 'local' optimal levels. With this analysis, we aim to better understand the spread of individual optimal ZOIs and what would be the best theoretically achievable RMS_ΔVLM. This analysis also provides a basis to motivate future investigations, in particular to identify systematic factors, which may lead to local different extents of the ZOI and to improve the accuracy of trend estimates.*

*Figure 5b displays the distribution of local optimal thresholds for TG-GNSS stations for the ALES-GESLA-ZOI dataset. Note that these estimates are not independent as they are based on prior knowledge of the ground truth VLM from GNSS. … An associated ideal selection of trends, based on optimal individual levels shown in Figure 5d would largely reduce to RMS_VLM to  0.89 mm/year. We emphasize that this constitutes the best RMS, which could theoretically be achieved with our dataset combination, if all of the local optimal levels could be systematically explained. This demonstrates that, albeit there might be room for minor improvements, there is still a strong limitation remaining in bringing the RMS below 1 mm/year.*

**Minor Issues:**

**The word 'frequent' is used throughout in the context: 'high frequent data'. I suggest this should be 'high frequency'.**

*Corrected*

**The terms accuracies and uncertainties are used throughout when singular is arguably more appropriate. E.g. ln 11 in the abstract.**

*We changed the terms to singular, when it is more appropriate.*

**Suggest a search and replace for units – some (e.g. km) do not have a space beforehand after the quantity.**

*Corrected, Note that for dataset names, we intentionally did not use space e.g. ALES-PSMSL-250km*

**L15 p1. Perhaps increasing instead of progressing? Perhaps : : :the spatial scales of the coherency in coastal sea level trends increase.**

*Both changed*

**L20, p1. : : :as the sea level rise signal itself: : :**

*Added*

**L21, p1. : : :can regionally account for a large fraction of the: : :**

*Corrected*

**L35, p2. I had expected TG instability to at least be mentioned here. I know it comes later, but consider a mention here.**

*In the paragraph the reviewer refers to, we have listed the possible causes of VLM and not the issues related to the kind of measurements. Therefore, we assume that with this comment the reviewer is referring to non-linear short-time-scale vertical movements which affect the TG records and would be seen as instabilities when trying to estimate a linear relative sea level trend, just as when a record changes its reference datum. Therefore we added this sentence in L36:* 'These and other non-linear processes at very short time scales are particularly challenging in the estimation of a linear rate of long-term VLM from TGs, since they would appear as instabilities in the record (similarly to a change in datum).'

**L40, p2. "global" trends range 1 to 3 mm/yr. Do you really mean "global"? If you do then I feel this could be made more specific to a time period of interest as it feels too vague.**

*We changed L40:*

'Given that global absolute sea level trends during the altimeter era is in the order of 3 mm/year (3.1+-0.1 mm/year from 1995 to 2018 as reported in Cazenave et al., 2018) one prerequisite for VLM estimation is that associated trend uncertainties should be at least one order of magnitude lower than those subtle signals (Wöppelmann and Marcos, 2016).'

Cazenave, A., Palanisamy, H., and Ablain, M.: Contemporary sea level changes from satellite altimetry: What have we learned? What are the new challenges?, Advances in Space Research, 62, 1639 – 1653, https://doi.org/https://doi.org/10.1016/j.asr.2018.07.017, http://www.sciencedirect.com/science/article/pii/S0273117718305799, 2018.

**L54, pg2. : : :solutions against those using measurements: : :**

*Added*

**L59, pg3. "considerable" isn't the right word here, consider change.**

*Changed*  *to* *comparably*

**P3 – See main issue #1.**

*Please refer to answers to #1*

**L80, p3. "synergistic applications" isn't quite right, consider change.**

*We changed*  *to* *'their combined use'.*

**L87, p3. : : :further develop: : : instead of "carry on" the latest progress? Suggest characterisation and quantification instead of detection?**

*Changed* *carry on* *to* *'further develop'*
*Changed* *detection* *to* *'characterization and quantification'*

**L96, p4. "were achieved" c.f. what?**

*We changed this line to*
*'Using this dataset, WM16 also obtained the most precise VLM estimates … '*

**L108, p4. suggest: : : :however, they reported insignificant improvements of: : :**

*Adopted*

**L126, p4. no comma needed after processes.**

*deleted*

**P6 – See main issues 2 and 3.**

*Please refer to answers to these main issues #2 and #3*

**L190, p7. The PSMSL constitutes: : :**

*Corrected*

**L191, p7. : : :for most sea-level research.**

*Corrected*

**L222, p8. low latitude maybe ambiguous for some readers. Do you mean nearer to the equator or poles? I note you use high-latitude later.**

*We changed* 'in low-latitude regions' *to* 'nearer to the equator'; *Later we added* '… higher latitudes (i.e. towards the poles) …'

**L240, p8. I'm unsure that "matching" is the correct word. "Differences formed using" perhaps?**

*Changed* 'matching' *to* 'differences formed using …'

**L252, p9. : : :the capability to compare: : : no comma after altimetry.**

*Corrected*

**L264, p9. duplicate as**

*deleted*

**L310, p12. performance not performant**

*Corrected*

**L330, p14. "next to SLA correction" could be significantly improved.**

*We rephrased these line as follows:*
'  *The SLA computation is affected by the instrumental errors of the range estimation and of each of the geophysical corrections (Ablain et al 2009). Such errors, as well as the measurement error of the TG itself, show up as residuals in the differenced time series. Moreover, sea level dynamics that do not reflect the variability observed at the tide gauge locations also contribute to the SAT-TG differences.* Therefore, … '

*M. Ablain, A. Cazenave, G. Valladeau, S. Guinehut. A new assessment of the error budget of global mean sea level rate estimated by satellite altimetry over 1993-2008. Ocean Science, European Geosciences Union, 2009, 5 (2), pp.193-201.*

**L_340, p14. Some comment from the authors regarding whether they consider PL+WN more appropriate than GGM for example would be useful.**

*We added the following lines to the end of this paragraph L340:*
'*We emphasize that for individual regions other noise models could be more appropriate than the implemented PL+WN model and would thus yield more realistic uncertainty estimates. An advanced regional spectral analysis to identify the most suitable models is however beyond the scope of this study.*'

**L349, p14. needs "ellipsoid" inserted after TOPEX/Poseidon.**

*Inserted*

**Figure 3: The distribution is unavoidably clustered. Does residual VLM show any spatial patterns?**

*We observe a slight tendency of higher GNSS than SAT-TG trends in Northern Europe and North America as well as lower GNSS trends toward equatorial regions and Australia. We add the associated trends and geographical distributions in Figure R5 below. Note that the causes of the spatial pattern of trend differences in Figure R5 requires further investigations.*

[Figure]

*Figure R5: Trend differences of GNSS minus SAT-TG VLM [mm/year] for ALES-GESLA-ZOI (80th percentile) (a) Map of trend differences; (b and c) Trend differences as a function of absolute latitude and latitude. Solid lines represent locally weighted regressions (Cleveland et al., 1988).*

*We also discuss the possible (and neglected) influence of contemporary mass distribution on GNSS and SAT-TG trend differences in the discussion.*

*Second part of added discussion paragraph from L524:*

*'… Next to altimeter bias drift, nonlinear VLM from contemporary mass redistribution (CMR) changes were shown to cause differences between VLM_SAT-TG and VLM_GPS, due to the different time periods covered (e.g. Kleinherenbrink et al. (2018)). Using GRACE (Gravity Recovery and Climate Experiment) observations Frederikse et al. (2019) demonstrated that associated deformations can cause VLM trends in the order of 1 mm/year. Therefore, they introduced a new method to reduce VLM_GPS by GIA and CMR signals to minimize their associated induced extrapolation biases. Kleinherenbrink et al. (2018) incorporated nonlinear VLM from CMR to assess the corresponding trend*

*differences between VLM_SAT-TG and VLM_GPS. They exposed that VLM_SAT-TG estimates are lower than VLM_GPS in many parts of North America and Europe and higher in subtropical/tropical regions as well as Australia and New Zealand (refer to Figure 9 in Kleinherenbrink et al. (2018)). Because northerly regions, for instance, are affected by stronger recent uplift, GNSS observations which cover shorter and more recent time spans than satellite altimetry detect more positive trends. For a set of 155 TG-GNSS pairs, integration of these signals slightly reduced the median bias from -0.14 mm/year to -0.07 mm/year, but had no significant effect on RMS. Given that most of the TG-GNSS stations used in this study are located in Europe, North America and Australia, CMR might as well alleviate the negative trend bias of ALES-GESLA-ZOI. Therefore, extending the validation platform, not only by using other homogeneous GNSS observations, but also GRACE and GIA estimates would strongly support identification and mitigation of such systematic errors.'*

**L370, p15.is "also" needed?**

*L370: Deleted *

**L399 p16. Suggest improve "which stronger deviate".**

*Changed '… which stronger deviate … ' to '… with stronger deviations from the TG records …'*

**L402, p16. This is the opposite sign however: : :**

*Due to the change of Envisat data and thus a change in the bias of trend differences (of ALES-GESLA-250km) we deleted this line.*

**L435 p18. assuming issues with e.g. tide model errors, the method presented will likely have a optimum threshold that still enables sufficient spatial averaging to mitigate that systematic effect as much as possible. See main issue 2.**

*We agree with the reviewer that the benefit of our method is dependent on thresholds and that these can be optimized either globally (as we do in the paper) or locally (which is a challenge to which currently we can only answer "a posteriori" using the comparison with the "truth" at the GNSS stations, as shown in Figure 5). In the text we highlight the influence of the sampling size on the global optimal threshold, because at very high levels the sample size decreases. We agree that there are many various factors which additionally contribute to these optimal levels (such as tidal model errors). For the discussion concerning the main issue 2, the reviewer is referred above to the corresponding answer.*

We add *from L429*:

*'RMS_VLM and trend uncertainties level off at very high thresholds and ultimately increase when only 5% of the data is used (Figure 5a and 5c). We argue that this is mainly related to a decrease in sampling-density of the time series included in the selection: At the 95$^{th}$ percentile, the median sample size (i.e. number of monthly averages in a time series) is 20% smaller that the sample size at the 80th percentile. Robust trend estimates require a minimum of samples, hence, using a reduced number of along-track data time series, even when they show a maximum degree of comparability, yields on a global average decreased trend accuracies (RMS_VLM).  We thus argue that the optimum threshold identified at about the 80th percentile (of the data sorted by RMS) represents a compromise between data-comparability, as well as sampling-density of altimetry data. We emphasize that there are numerous factors, other than the time period covered, which may contribute to a lack of comparability of SAT-TG and GNSS trends. We further elaborate those in the subsequent discussion section 5.'*

**L439, p18. Is bisector the correct term? 1:1 line?**

*Changed to* *1:1 line*

**L441, p18. Unclear what you mean by outstrips improvements induced by: : :. Clarify.**

*We changed the sentence:*

*This result proves the importance of using such a refined selection procedure (ZOI), as this approach outstrips the improvements induced by the different altimeter or tide gauge data combinations.*

*To*

*'These results underpin that a refined selection procedure (ZOI) represents the dominant advancement, as this approach outstrips the improvements (in terms of trend accuracy and uncertainty) which are obtained from using different altimeter or tide gauge data combinations.'*

**Figure 4 – Comment error bars are 1 sigma?**

*The error-bars are 1 sigma, we added that to the figure caption.*

**L447, p20. The residual annual cycle criterion: : :**

*corrected*

**L450, p20. I tend to agree with the statement, but what about the bias in the accuracy as arguably the low frequency component may lead to larger biases in deltaVLM (as well as scatter). A comment maybe useful here.**

[Figure]

*Figure R6: Shown is the evolution of the bias (median of trend differences) depending on different criteria and thresholds for ALES-GESLA-ZOI (as in Figure 5)*

*We give some additional information on the evolution of the trend bias in Figure R6 above. We do not find a similar convergence of the biases to a global optimum as for accuracy or uncertainty. The values exhibit fluctuations in the order of 0.1 mm/year. There is also no significant difference between using different criteria. Thus, we add to L450:*

*'Considering improvements in the bias of trend differences, we find no significant differences in using different thresholds. In contrast to the improvements in accuracy (as shown in Figure 5), the median*

*ΔVLM does not converge to a global optimum. Therefore, we discuss the contribution of other factors affecting the comparability of SAT and GNSS in section 5.2'*

**Figure 5. In the legend, unclear what AM is. I assume something to do with Annual, but needs clarification in the caption.**

*We changed AM to AC, and added a description in the caption*

**L455, p22. estimates do not always**

*Added*

**L458, p22. See main issue #1.**

*As given in the response to main issue #1 we added a section discussing systematic errors from L524.*

**L460, p22. Is average the best word here. I understand your point, but statistically speaking? Perhaps denote metrics of performances derived from the global dataset?**

*We changed 'The results presented in Figure 5 and Table 2 denote average performances for the globally distributed TG-GNSS station pairs for ALES-GESLA-ZOI and support …' to*

*'The results presented in Figure 5 and Table 2 denote metrics and performances derived from the global TG-GNSS dataset for ALES-GESLA-ZOI …'*

**L463, p22. at every coastal site considered.**

*Changed*

**L469, p22. "large" in inadequately defended. See issue #3.**

*Changed. See previous response to #3*

**Section 5.1 was challenging to read, Fig 4b and d were difficult to interpret. I suggest some further elaboration.**

*Please refer to the changes made in response to a previous comment (improvements of section 5.1.)*

**L494, p23, no comma needed after we note.**

*deleted*

**L503, p24. I understand your point here but I feel more emphasis should be placed on the sensitivity of the ALT and TG to the same phenomena. If both were sensing 100% the same signal, the time span is not as critical. Again, I come back to main issue #1 here.**

*In this chapter, we investigated how the accuracy of the estimates changed with different time series lengths. We found a dependence of an optimal spatial selection of SLAs on time scale. This dependency indicated that with increasing time scales also the spatial scales, which would yield the best accuracy of the estimates, would increase. We argued that with increasing record lengths SL-trends would emerge from the shorter term variability and also be similar to TG trends at greater distance.*

*As far as we understand, the reviewer points to the possible influence of systematic drift biases in causing overall biases when comparing ALT-TG and GNSS trend estimates. We agree that such an effect would be less time scale dependent. The finding that accuracy does not improve as strongly as uncertainty could also indicate limitations set by systematic errors such as altimeter drift biases.*

*In our understanding, in the context of the analysis presented in the chapter, a drift bias would cause a constant offset in the trend residuals and would thus not influence the time-scale dependencies found in this analysis.*

*However, given that the bias itself reduces the accuracy of our estimates by about 10% (when simply comparing the STD of the trend differences against the RMS), we further elaborated associated issues at several positions in the text. (refer to previous responses, e.g. L385 and L524 )*

**L535, p26. Quantify "much of" and "close vicinity to the coast". See main issue 3.**

*Quantified, see response to #3*

**L538, p26. : : :not be unequivocally**

*corrected*

**L560, p26. Are these statistics medians?**

*Yes, we added 'median formal uncertainties …'*

**L565, p26. : : :and dedicated coastal altimetry: : : confine ZOIs and increase: : : : : :the global coastline which improved uncertainty.**

*Corrected (we assume it should be 'with improved uncertainty')*

---

## Author Response (AR2)

Review response #2

To Christopher Watson, November, 3 2020

We are very glad about the approval of our manuscript. We really appreciate all your help with the manuscript revision, which not only aided us to much better identify important (and so far neglected) issues concerning the limitations of our approach, but also were a vital contribution to enhance our line of argument and readability of the manuscript.

We are also very grateful for the support of the editor Joanne Williams throughout the submission process.

Sincerely,

The authors

p3 l83: Strictly speaking, the issues with TOPEX were global, but undoubtedly had a regional expression. Suggest: ...can be affected by globally and regionally varying drifts...

**A: changed to 'can be affected by globally and regionally varying drifts'**

p3 l86: The additions to this paragraph are an improvement - however, they are not explicit in saying that the technique does not consider estimation of the drift/systematic error issue w.r.t the reference missions. I suggest that a statement to this effect is added here, and/or in the paragraph around line 142 on page 5. The additional words in the last paragraph on page 6 partly addresses this, however this issue should be clear from the introduction.

We added in p3 l86: 'Conversely, SAT-TG VLM estimates are affected by the altimeter drift or by errors originating from the intermission drift biases (or drifts w.r.t the reference mission). '

p235 l8: The addition of this paragraph clarifies the treatment of tides nicely - I do however remain concerned regarding the aliased sampling of the residual longer period tides at TG sites. At some sites I would consider this a significant contribution to the error budget and it deserves mention in the discussion as a limiting factor. I did not see that this issue was mentioned in the discussion - I would consider this a useful addition.

We fully agree that the residual longer period tides contribute further to the residuals in SAT minus TG time series and should be accounted for in future studies. We mentioned this factor in a new summarizing paragraph at the end of the discussion:

*'... Hence, we maintain the main advantage of using tide gauges for VLM estimation: The large global distribution compared to continuous GNSS-measurements.*

We highlight that the SAT-TG estimates are not only limited by the broad spectrum of error sources, ranging from systematic to correction errors, such as the residual long period tides remaining in the TG time series, which all contribute to the error budget of the estimates. Another factor is the possible nonlinearity of the VLM itself, which strongly hampers the comparability with measurements from other geodetic techniques, when sampled over different time spans. Thus, addressing this issue in SAT-TG time series could represent a further crucial improvement of the application.' p15 I357: I understand the sentiment, however the sentence commencing with "Moreover" could perhaps be improved. Perhaps: Moreover, sea level dynamics that are not common between the TG and altimeter observation locations will also contribute to the SAT-TG differences."

**Corrected!**

p25, l522: Perhaps: ...which may lead to locally different extents...

**Corrected**

p29, I613: Rather than "detect more positive trends" perhaps mention sampling regions where significant GIA exists with a substantial uplift signal...

Here refer to effects associated with contemporary mass redistribution. We add an example of a region, which is particularly affected by contemporary mass changes (due to recent ice melt and elastic deformations):

'Because northerly regions like North America, are affected by dynamic changes in CMR, GNSS observations which cover shorter and more recent time spans than satellite altimetry detect stronger uplift signals.'

p29 section 5.2: This is a useful addition to the manuscript. Around line 602, "but does not feature a calibration against TG" could be expanded to clarify the role of MMXO, e.g.: ...but does not feature a calibration against TGs, nor can resolve regionally coherent systematic errors in the trend within the reference missions. (if this is indeed the case? - I guess I am seeking clarification of what the MMXO can and can not do w.r.t. regionally coherent signals in e.g. Jason-2 etc.

The MMXO indeed reduces regionally coherent systematic errors, which is not clearly mentioned here (we adressed this briefly in section 2.1.). Therefore, we change the sentence to:

'Moreover, as mentioned before, the multi-mission calibration applied (MMXO) reduces intermission biases as well as regionally coherent systematic errors, but does not feature a calibration against TG.'

**Very minor points: p2 I44: this instead of those?**

**Yes! Corrected**

Throughout: Some of the brackets in references seem inconsistent - copy editing should resolve this.

Corrected

**Throughout: Inconsistent use of TG and tide gauge. Suggest find and replace?**

Corrected

[revised manuscript text omitted]